# NAD+ repletion with niacin counteracts cancer cachexia

Marc Beltrà [1,14], Noora Pöllänen [2,14], Claudia Fornelli[1], Kialiina Tonttila[3,4], Myriam Y. Hsu[5], Sandra Zampieri[6,7,8], Lucia Moletta[6], Samantha Corrà[9], Paolo E. Porporato [5], Riikka Kivelä [3,4,10], Carlo Viscomi [7,11], Marco Sandri [7,9], Juha J. Hulmi [4], Roberta Sartori [7,9] ✉, Eija Pirinen [2,12,13] ✉ & Fabio Penna [1] ✉

Cachexia is a debilitating wasting syndrome and highly prevalent comorbidity in cancer patients. It manifests especially with energy and mitochondrial metabolism aberrations that promote tissue wasting. We recently identified nicotinamide adenine dinucleotide (NAD+) loss to associate with muscle mitochondrial dysfunction in cancer hosts. In this study we confirm that depletion of NAD+ and downregulation of *Nrk2*, an NAD+ biosynthetic enzyme, are common features of severe cachexia in different mouse models. Testing NAD+ repletion therapy in cachectic mice reveals that NAD+ precursor, vitamin B3 niacin, efficiently corrects tissue NAD+ levels, improves mitochondrial metabolism and ameliorates cancer- and chemotherapy-induced cachexia. In a clinical setting, we show that muscle *NRK2* is downregulated in cancer patients. The low expression of *NRK2* correlates with metabolic abnormalities under-scoring the significance of NAD+ in the pathophysiology of human cancer cachexia. Overall, our results propose NAD+ metabolism as a therapy target for cachectic cancer patients.

Cancer cachexia (CC) is a complex multifactorial syndrome resulting from both tumor-induced host adaptation and anti-cancer treatment side effects, being present in and worsening the outcome of more than half of all cancer patients. CC is clinically characterized by an involuntary loss of body weight mainly due to muscle wasting, with or without depletion of adipose tissue that impair patients' quality of life and survival[1]. Several energy metabolic abnormalities including mitochondrial dysfunction have been characterized in CC suggesting that targeting energy metabolism could be useful when designing novel anti-cachexia treatments[2,3]. Recently, our study on C26 adenocarcinoma bearing mice showed that the decline of mitochondrial oxidative phosphorylation (OXPHOS) occurred in parallel with depleted NAD+ levels in the skeletal muscle[4]. Given that NAD+ is an essential cofactor for multiple mitochondrial redox reactions, alterations of its levels can affect mitochondrial homeostasis and subsequently tissue function. In support of this notion, mRNA levels of genes related to NAD+

[1]Experimental Medicine and Clinical Pathology Unit, Department of Clinical and Biological Sciences, University of Torino, Turin, Italy. [2]Research Program for Clinical and Molecular Metabolism, Faculty of Medicine, University of Helsinki, Helsinki, Finland. [3]Stem Cells and Metabolism Research Program, Faculty of Medicine, University of Helsinki, Helsinki, Finland. [4]Faculty of Sport and Health Sciences, NeuroMuscular Research Center, University of Jyväskylä, Jyväskylä, Finland. [5]Department of Molecular Biotechnology and Health Sciences, Molecular Biotechnology Center, University of Torino, Turin, Italy. [6]Department of Surgery, Oncology and Gastroenterology, University of Padova, Padova, Italy. [7]Department of Biomedical Sciences, University of Padova, Padova, Italy. [8]CIR-MYO Myology Center, University of Padova, Padova, Italy. [9]Veneto Institute of Molecular Medicine, Padova, Italy. [10]Wihuri Research Institute, Helsinki, Finland. [11]Study Centre for Neurodegeneration, University of Padova (CESNE), Padova, Italy. [12]Research Unit of Biomedicine and Internal Medicine, Faculty of Medicine, University of Oulu, Oulu, Finland. [13]Medical Research Center Oulu, Oulu University Hospital and University of Oulu, Oulu, Finland. [14]These authors contributed equally: Marc Beltrà, Noora Pöllänen. ✉e-mail: roberta.sartori@unipd.it; eija.pirinen@helsinki.fi; fabio.penna@unito.it

biosynthesis positively correlate with the expression of genes regulating muscle mitochondrial biogenesis, muscle mass growth and muscle regeneration in mice[5,6]. Recent rodent and human studies have reported NAD+ depletion as a pathological hallmark for various muscle diseases including sarcopenia and mitochondrial myopathy[6–8]. NAD+ depletion is typically caused by impaired NAD+ biosynthesis, increased activities of NAD+ degrading enzymes, changes in metabolic reactions relying on NAD+/NADH redox couple or a combination of all these alterations. In our above-mentioned work in C26-bearing mice, skeletal muscle NAD+ loss was associated with a strong transcriptional downregulation of the NAD+ biosynthetic enzyme *nicotinamide riboside kinase 2* (*Nrk2*). This salvage pathway enzyme metabolizes vitamin B3, nicotinamide riboside (NR), toward NAD+ and is regulated by stress and alterations in the intracellular NAD+ and energy supply[9]. NR has been previously published to alleviate cachexia in mice bearing the C26 tumor[10], although the demonstration of NAD+ replenishment was lacking. Currently, NR has limitations of use in clinical practice, as only our long-term clinical study[11] has demonstrated positive outcomes on muscle mitochondrial metabolism compared to other published short-term trials[12,13]. In contrast, the other vitamin B3 form, niacin (NA), has a proven safety record in humans and it has been published to improve muscle mitochondrial function and muscle strength in patients with adult-onset mitochondrial myopathy[14]. Despite previous findings, unsolved questions remain: (1) how common muscle NAD+ depletion and *Nrk2* downregulation are in CC induced by distinct tumors, (2) is NAD+ metabolism disturbed in other tissues beyond the skeletal muscle, and (3) can NAD+ repletion with NA mitigate the symptoms of cancer cachexia.

In this study, we performed a wider and deeper characterization of NAD+ and energy metabolism impairments in CC and examined the potential therapeutic role of NA supplementation. Skeletal muscle NAD+ deficiency was detected in mice with severe CC, whereas muscle *Nrk2* loss was observed in several preclinical CC models and was

validated in cancer patients. In addition, acute and chronic CC in mice resulted in depletion of all the NAD metabolites in the liver. NA corrected NAD+ deficiency and increased mitochondrial biogenesis in both skeletal muscle and liver of cachectic mice, partially restoring muscle mass loss and energy metabolism changes. Thus, our findings propose that *Nrk2* repression and NAD+ metabolism aberrations are prevalent features of CC. Correcting NAD+ metabolism has a protective role in maintaining adequate energy homeostasis and preventing cachexia in tumor-bearing animals.

## Results

### Disturbed muscle NAD+ metabolism is prevalent in CC

In this study, muscle NAD+ metabolism was investigated in three different CC experimental settings using either female or male mice in the distinct models; however, a sexual dimorphism cannot be excluded. Out of the three models, body weight was significantly reduced only in C26-bearing mice treated with Folfox chemotherapy (C26-F mice) while decreased muscle mass was a typical feature for all models (Supplementary Fig. 1a–d). C26-F mice exhibited an exacerbation of NAD+ depletion (Fig. 1a) in comparison to our previous publication in chemotherapy-naïve C26-bearing ones (NAD+ decrease: 30.8% vs 12.5%, respectively)[4]. Consistently, a reduction in NAD+ was observed also in the skeletal muscle of KPC tumor-bearing mice (Fig. 1a), a CC model representative of pancreatic ductal adenocarcinoma[15]. In order to better model CC, we assessed NAD+ content in the skeletal muscle of Villin-Cre/Msh2loxP/loxP (VCM) mice that slowly and spontaneously develop neoplasms due to the conditional knock-out of the mismatch repair gene Msh2 in the enterocytes of the intestinal mucosa[16]. Although presenting with significant splenomegaly and anemia, body weight loss and muscle wasting due to tumor progression are moderate in the VCM model compared to age-matched Msh2loxP/loxP mice (controls; Supplementary Fig. 1d–f), supporting the idea of a milder and more chronic model of CC. As opposed to the previous severe CC

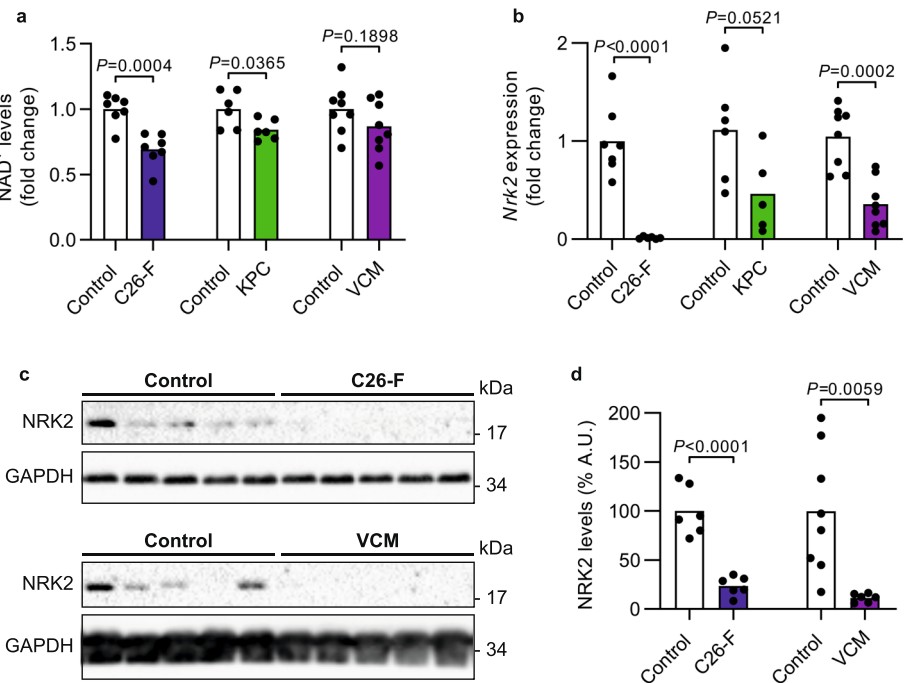

**Fig. 1 | NAD+ depletion and *Nrk2* downregulation as hallmarks of cancer cachexia in the skeletal muscle. a** NAD+ levels in the skeletal muscle of C26-F (*n* = 7 per group), KPC (*n* = 6 per group) and VCM (*n* = 8 per group) CC models. **b** Expression of the *Nrk2* gene in the skeletal muscle of C26-F (controls *n* = 7, C26-F *n* = 6), KPC (controls *n* = 6, KPC *n* = 5) and VCM (*n* = 8 per group) mice. Data are normalized to housekeeping gene expression and displayed as fold change.

**c, d** Representative western blotting bands and densitometry analysis of NRK2 protein levels on C26-F (*n* = 6 per group) and VCM (controls *n* = 8, VCM *n* = 6) mice. GAPDH was used as loading control. Mice age: C26-F 7-month-old, KPC 3-month-old, VCM 13-month-old. All controls are age-matched. Data are shown as means with individual values. Statistical analysis was performed using two-tailed Student's *t* test. Original raw data are provided as a Source Data file. A.U. arbitrary units.

models, skeletal muscle NAD$^+$ levels were not significantly reduced in VCM mice (Fig. 1a). In chemotherapy-naive C26 tumor-bearing mice, decreased NAD$^+$ levels associated with the repression of the NAD$^+$ biosynthetic gene *Nrk2*[4]. Consistently, cachectic mice showed downregulation of muscle *Nrk2* gene expression in both C26-F and VCM models compared to controls, while a trend toward *Nrk2* reduction was detected in KPC mice (Fig. 1b). Remarkably, muscle NRK2 protein levels were nearly undetectable in C26-F and VCM mice when compared to their respective control group (Fig. 1c,d). Altogether, these data demonstrate that skeletal muscle NAD$^+$ content is depleted in severe CC and that *Nrk2* downregulation is a common feature of experimental CC independently from the severity of the syndrome.

## Low *NRK2* associates with muscle metabolome alterations in human CC

To validate the preclinical data in humans, *NRK2* gene expression was assessed in skeletal muscle biopsies from colorectal and pancreatic cancer patients and compared to healthy subjects. The samples originate, with some additions, from a previous study[17] (patient characteristics are summarized in Supplementary Table 1). *NRK2* gene expression decreased in weight-stable cancer patients and showed a trend toward further downregulation in cachectic patients compared to weight-stable ones (Fig. 2a) confirming that *NRK2* loss is a novel common alteration in CC. To examine the relationship between *NRK2* loss and muscle metabolism in CC, we selected 10 patients with the highest (comparable to healthy controls) and 10 with the lowest (almost tenfold decrease) *NRK2* gene expression that did not differ in terms of body weight loss (Fig. 2b,c). Moreover, *NRK2* gene expression levels were independent of muscle mass and wasting, as no association was found with macroscopic clinical features of the current cachexia diagnostic criteria, mainly based on body weight loss and sarcopenia (Supplementary Table 2). The metabolomic characterization (Supplementary Data 1) revealed a peculiar signature in the low *NRK2* expressing muscles that clearly distinguished from high *NRK2* and healthy counterparts (Fig. 2d). Low *NRK2* muscles showed an accumulation of glycolysis intermediates (Fig. 2e) with fructose-6-phosphate (F6P) and citrate levels presenting an inverse correlation to *NRK2* gene expression (Fig. 2f,g). In addition, other metabolites including nucleotides and amino acids accumulated in low *NRK2* muscles (Fig. 2h–j; Supplementary Fig. 2a–j), representing the result of impaired energy metabolism and potentially hypercatabolism. This trait can be partially observed also in the serum of the same individuals (Fig. 2k), suggesting that muscle energy failure could be diagnosed with minimally invasive procedures.

## Niacin rescues muscle NAD$^+$ levels and ameliorates CC

NRKs catalyze the utilization of NAD$^+$ precursor NR via the salvage pathway. In the skeletal muscle, *Nrk2* is the most expressed isoform in both BALB/c and C57BL/6 mouse strains as compared to *Nrk1* (Supplementary Fig. 3a). Considering the *Nrk2* repression in CC and the lack of a simple, translatable tool to correct *Nrk2* expression, we decided to use the NAD$^+$ booster NA, a precursor that is utilized for NAD$^+$ biosynthesis through the Preiss-Handler pathway thus bypassing NRK2[18]. C26-F animals were treated with a daily dose of NA (150 mg/kg) starting from day 4 after C26 implantation until day 28 (Fig. 3a). In addition to NAD$^+$ depletion (Fig. 1a), C26-F mice presented with a significant decrease of NADH and NADPH levels while NADP$^+$ levels were similar in comparison to controls (Supplementary Fig. 3b). Besides *Nrk2* loss, C26-F mice showed an overall repression of genes involved in NAD$^+$ biosynthesis via the salvage and Preiss-Handler pathways (Supplementary Fig. 3c) and enhanced enzyme activity of poly(ADP-ribose) polymerases (PARPs), one of the main consumers of cellular NAD$^+$ pool operating, for example, in DNA repair (Supplementary Fig. 3d). Interestingly, NA increased skeletal muscle NAD$^+$ and NADP$^+$ concentrations almost to the control levels and slightly impacted on NADH and

NADPH levels (Fig. 3b, Supplementary Fig. 3b). Moreover, NA supplementation improved cachexia symptoms by counteracting the loss of body weight and muscle mass and partially rescuing grasping strength (Fig. 3c–e, Supplementary Fig. 3e). Consistent with our previous report[19], C26-F mice presented with decreased skeletal muscle protein synthesis, increased ratio of the active LC3B isoform (LC3B-II; Fig. 3f–h) and AMPK$^{Thr172}$ phosphorylation (Fig. 3f–i), suggestive of increased autophagy and energy shortage, respectively. Interestingly, both protein synthesis and LC3B-II accumulation were in part rescued by NA (Fig. 3f–h), while AMPK activation was partially prevented (Fig. 3f,i).

Based on the in vitro experiments, NA increased C26 cell proliferation, whereas administration of oxaliplatin or 5-fluorouracil (components of Folfox cocktail) neutralized this effect (Supplementary Fig. 3f). In addition, NA increased the number of dead cells and partially potentiated oxaliplatin toxicity (Supplementary Fig. 3g). In vivo, tumor mass was not significantly affected by NA in C26-F mice (Supplementary Fig. 3h) although a trend toward reduction was observed. Dissecting whether NA action directly impinges on the tumor, we found that tumors from both treated and untreated animals presented similar levels of *Il6* and *Inhba* transcripts (Supplementary Fig. 3i), indicative of unaffected cytokine production by the tumor. In addition, treated and non-treated mice showed a similar liver gene expression profile of the systemic inflammation markers *Il6*, *Inhba* and *Saa1-2* (Supplementary Fig. 3j). Consistently, no improvement in animal survival was observed when the C26-F experimental protocol was extended to 49 days (Supplementary Fig. 3k), suggesting that improved cachexia by NA is not impacting on tumor progression leading to animal death.

To explore the impact of NA in a more chronic and clinically relevant model of CC presenting with *Nrk2* loss, VCM mice were treated with NA for 28 days at 12 months of age to allow the tumors to spontaneously develop (Fig. 3j). One limitation of this study, considering the heterogeneity of spontaneously developed tumors, as normally happening in humans, is the inability to dissect the impact of NA on tumor burden. In line with the preserved NAD$^+$ content in VCM mice (Fig. 1a), NAD metabolites, the expression of NAD$^+$ biosynthetic genes and PARP activity remained unaltered between control and VCM groups (Supplementary Fig. 4a–c). NA supplementation had a minimal impact on skeletal muscle NAD$^+$ and other NAD metabolites (Fig. 3k, Supplementary Fig. 4a). Nonetheless, NA protected VCM mice from muscle mass loss (Fig. 3l–n), preventing muscle autophagosome accumulation (Fig. 3o,p) and increased expression of E3 ubiquitin ligases, autophagy and mitophagy genes (Supplementary Fig. 4d–f) without blunting the induction of the hepatic inflammatory markers *Il6*, *Inhba* and *Saa1-2* (Supplementary Fig. 4g). Overall, NA showed beneficial effects on CC by preventing muscle loss and induction of autophagy markers in both severe and mild CC models.

## Niacin improves muscle mitochondrial biogenesis in experimental CC

Muscle mitochondrial dysfunction has gained importance in recent years as a crucial feature of CC[20]. In the current experimental setting, mitochondrial respiration was significantly decreased in the skeletal muscle of C26-F mice compared to healthy controls (Fig. 4a), while biochemically significant changes in mitochondrial respiratory chain complex I-IV enzymatic activities were not detected (Supplementary Fig. 4h–k). The defective mitochondrial respiration was associated with drastically lowered skeletal muscle ATP content (Fig. 4b) and a significant decline in the markers for mitochondrial content; mitochondrial DNA (mtDNA) amount (Fig. 4c) and citrate synthase activity (Fig. 4d). In addition, C26-F mice presented with reduced protein expression of several subunits of the mitochondrial oxidative phosphorylation (OXPHOS) complexes (Fig. 4e,f), the mitochondrial mass marker TOMM20 and the main activator of mitochondrial biogenesis

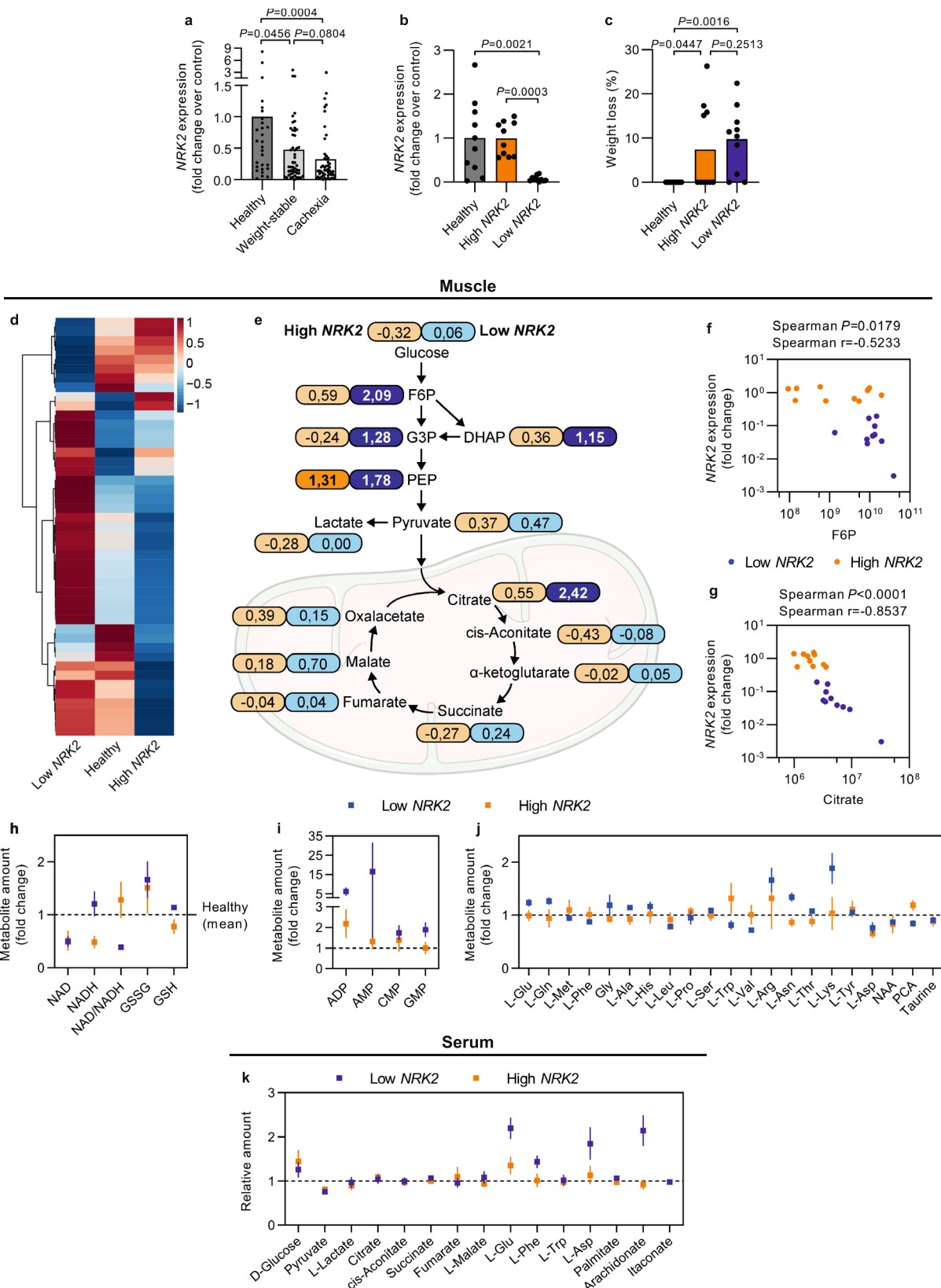

PGC-1α (Fig. 4e,g,h). In addition, several genes involved in mitochondrial biogenesis were downregulated (Supplementary Fig. 4l), whereas PINK1, a marker of mitochondrial damage and depolarization[21], accumulated in the skeletal muscle of C26-F mice (Fig. 4e,i). NA supplementation did not significantly influence mitochondrial respiration (Fig. 4a) but a trend for increased complex II enzymatic activity was observed (Supplementary Fig. 4i). This was translated into a partial rescue of intracellular ATP content (Fig. 4b). In addition, NA improved mtDNA amount, citrate synthase activity (Fig. 4c,d) and expression of OXPHOS complex subunits, as well as TOMM20 and PGC-1α protein levels (Fig. 4e–h) without impacting on PINK1 accumulation (Fig. 4e,i). No transcriptional induction of genes crucial for mitochondrial

**Fig. 2 | Metabolome analysis in human muscle biopsies and serum according to *NRK2* expression. a** *NRK2* gene expression in *rectus abdominis* muscle biopsies from healthy volunteers ($n = 28$) and cancer patients stratified as weight-stable ($n = 49$) or cachectic ($n = 53$). **b** *NRK2* gene expression in healthy volunteers ($n = 10$; gray), high *NRK2*-expressing ($n = 10$; orange) and low *NRK2*-expressing ($n = 10$; blue) groups. Data are normalized to housekeeping gene expression and displayed as fold change over control group. **c** Weight loss percentage in the previous 6 months before muscle biopsy and serum collections from patients with high or low *NRK2* expression that were selected for metabolome analysis. **d** Heatmap highlighting the abundance of all the detected metabolites in the skeletal muscle. Colors represent the mean *Z* score of each group. **e** Schematic illustration presenting the relative abundance of metabolites categorized in the "glycolysis" and the "Kreb's Cycle"

pathways. Numbers in the boxes represent log2(fold change) of either high *NRK2* (orange) or low *NRK2* (blue) groups compared to healthy volunteers for each metabolite. **f,g** Correlation plot between *NRK2* gene expression and F6P (**f**) or citrate (**g**) abundance in individual cancer patients ($n = 20$). **h,i** Relative amount of metabolites detected in the skeletal muscle categorized as (**h**) "redox", (**i**) "nucleotides", and (**j**) "amino acids". **k** Relative amount of detectable metabolites in the serum. Data display: (**a**–**c**) are fold change means with individual values, (**e**) are fold change vs healthy group, (**h**–**k**) are fold change ± SEM vs healthy group (dotted line). Statistical analysis: (**a**–**c**) two-tailed Kruskal–Wallis test with adjustment for multiple testing (Benjamini, Krieger and Yekuteli), (**f,g**) two-tailed Spearman's correlation. Original raw data are provided as a Source Data file. PCA pyroglutamate, NAA N-acetyl aspartate.

biogenesis was observed upon NA (Supplementary Fig. 4l) indicating that NA's effect on mitochondrial metabolism is mediated via post-transcriptional mechanisms.

In the skeletal muscle of VCM mice, an impairment of mitochondrial respiration was also observed (Fig. 4j). Contrary to the C26-F model, ATP content and mtDNA abundance was similar between VCM and control mice (Fig. 4k,l). However, VCM mice showed a decrease in the muscle protein content of OXPHOS complex subunits MTCO1 (CIV) and SDHB (CII) (Fig. 4m,n), as well as in TOMM20 (Fig. 4m,o) and PGC-1α (Fig. 4m,p), when compared to controls. PINK1 did not accumulate into the VCM muscle (Fig. 4m,q). NA treatment had no influence on muscle mitochondrial respiration and ATP content (Fig. 4j,k), but it increased mtDNA amount (Fig. 4l), and rescued PGC-1α levels (Fig. 4m,p). Similarly to C26-F mice, VCM mice showed downregulation of genes promoting mitochondrial biogenesis, although in this case NA was sufficient to correct the expression of *Errα* (Supplementary Fig. 4m). Overall, NA treatment modestly improves skeletal muscle mitochondrial status in two distinct models of experimental CC.

## Niacin corrects hepatic NAD⁺ deficiency and mitochondrial alterations

CC is a complex metabolic syndrome in which the liver has a crucial role in the control of systemic energy and glucose metabolism[2]. As the liver also contributes to the systemic regulation of NAD⁺ synthesis and recycling[22], we examined how CC and NA treatment influence liver condition and NAD⁺ metabolism in C26F and VCM mice.

C26F mice showed liver hypertrophy with depleted hepatic glycogen and total glutathione levels (Supplementary Fig. 5a–d), together with a severe decline in blood glucose levels (Supplementary Fig. 5e). NA supplementation slightly improved total glutathione content and glycemia whereas NA had no significant effects on liver size or hepatic glycogen levels (Supplementary Fig. 5a–e). All hepatic NAD metabolites (NAD⁺, NADH, NADP⁺ and NADPH) were markedly reduced in C26-F mice as compared to controls (Fig. 5a). NAD⁺ depletion likely originated from a strong downregulation of NAD⁺ biosynthetic enzymes of salvage and Preiss-Handler pathways, including the liver isoform of nicotinamide riboside kinase *Nrk1* (Supplementary Fig. 5f), not from enhanced NAD⁺ consumption via PARPs (Supplementary Fig. 5g). NA restored all hepatic NAD metabolite concentrations in C26-F mice (Fig. 5a). The livers of C26-F mice showed increased protein synthesis as opposed to skeletal muscle, even if autophagosome (LC3B-II) accumulation was similarly increased (Supplementary Fig. 5h–j). Neither protein synthesis nor LC3B levels were influenced by NA. Even though hepatic protein synthesis, one of the most energy consuming processes in the cell, was significantly enhanced, mitochondrial respiration only showed a trend toward increase while mtDNA abundance did not significantly change in the liver of C26-F mice (Fig. 5b,c). To further aggravate the energy metabolism scenario, citrate synthase activity (Fig. 5d) and protein expression of OXPHOS complex subunits and TOMM20 were decreased in C26-F mice as compared to controls (Fig. 5e–g). Similarly to skeletal muscle, a reduction of transcripts was

observed for the activators of mitochondrial biogenesis *Tfam* and *Errα* in C26-F mice (Supplementary Fig. 5k). NA did not affect liver mitochondrial respiratory capacity (Fig. 5b). However, NA increased mtDNA amount above control levels and partially rescued the protein content of the respiratory complexes II, III and IV as well as TOMM20 (Fig. 5c–g). No transcriptional induction of mitochondrial biogenesis markers occurred in the liver after NA supplementation (Supplementary Fig. 5k).

As in C26-F mice, VCM mice showed hepatomegaly (Supplementary Fig. 6a) and a dramatic reduction in hepatic NAD⁺, NADH, NADP⁺ and NADPH content as compared to controls (Fig. 5h). The expression of NAD⁺ biosynthetic genes and PARP activity in the liver remained fairly stable (Supplementary Fig. 6b,c). While not affecting liver size, NA supplementation restored hepatic NAD metabolite concentrations in VCM mice (Fig. 5h, Supplementary Fig. 6a). Differently from the C26-F model, mitochondrial respiration was significantly reduced in the livers of VCM mice (Fig. 5i). Hepatic mtDNA amount did not differ between VCM and control mice (Fig. 5j). Yet, protein levels of OXPHOS complex III, IV and V subunits were significantly decreased in tumor-bearing animals (Fig. 5k,l). The protein expression of TOMM20 was similar in VCM and control mice (Fig. 5k,m). NA supplementation normalized mitochondrial respiration to control levels (Fig. 5i) and increased mtDNA amount and ATP5 (CV subunit) expression as compared to non-treated VCM mice (Fig. 5j,l). The transcription of mitochondrial biogenesis markers partially decreased in VCM mice and NA only improved the expression of *Tfam* (Supplementary Fig. 6d). In conclusion, these findings reveal that CC is characterized by hepatic NAD metabolite deficiency and mitochondrial abnormalities that are partially restored boosting NAD⁺ metabolism with NA.

## Discussion

Disturbed skeletal muscle NAD⁺ metabolism has recently emerged as a molecular determinant of murine CC[4]. Our study reveals that the downregulation of muscle NAD⁺ biosynthetic enzyme *NRK2* is a common feature of murine and human CC, allowing the identification of patients with metabolic disturbances. Importantly, rescuing NAD⁺ levels protects from cancer- and chemotherapy-induced muscle wasting in mice.

NAD⁺ depletion and perturbed NAD⁺ biosynthesis are well-established pathophysiological factors of diseases characterized by muscle mitochondrial dysfunction and disturbed energy metabolism, such as mitochondrial myopathies and sarcopenia[8,14]. As summarized in Fig. 6, muscle NAD⁺ depletion occurs mainly in mice with severe CC. In contrast, the downregulation of *Nrk2* was detected in the skeletal muscle of all mouse models including the milder and chronic VCM model of CC even without NAD⁺ depletion. This finding suggests that *Nrk2* loss may precede the development of NAD⁺ metabolism disturbances and muscle loss, similarly to the situation in chemotherapy-induced cachexia[23]. Consistently, we show that cancer patients exhibit muscle *NRK2* repression (Fig. 6), already in weight-stable condition and exacerbated in overt cachexia. As muscle *NRK2* loss occurred

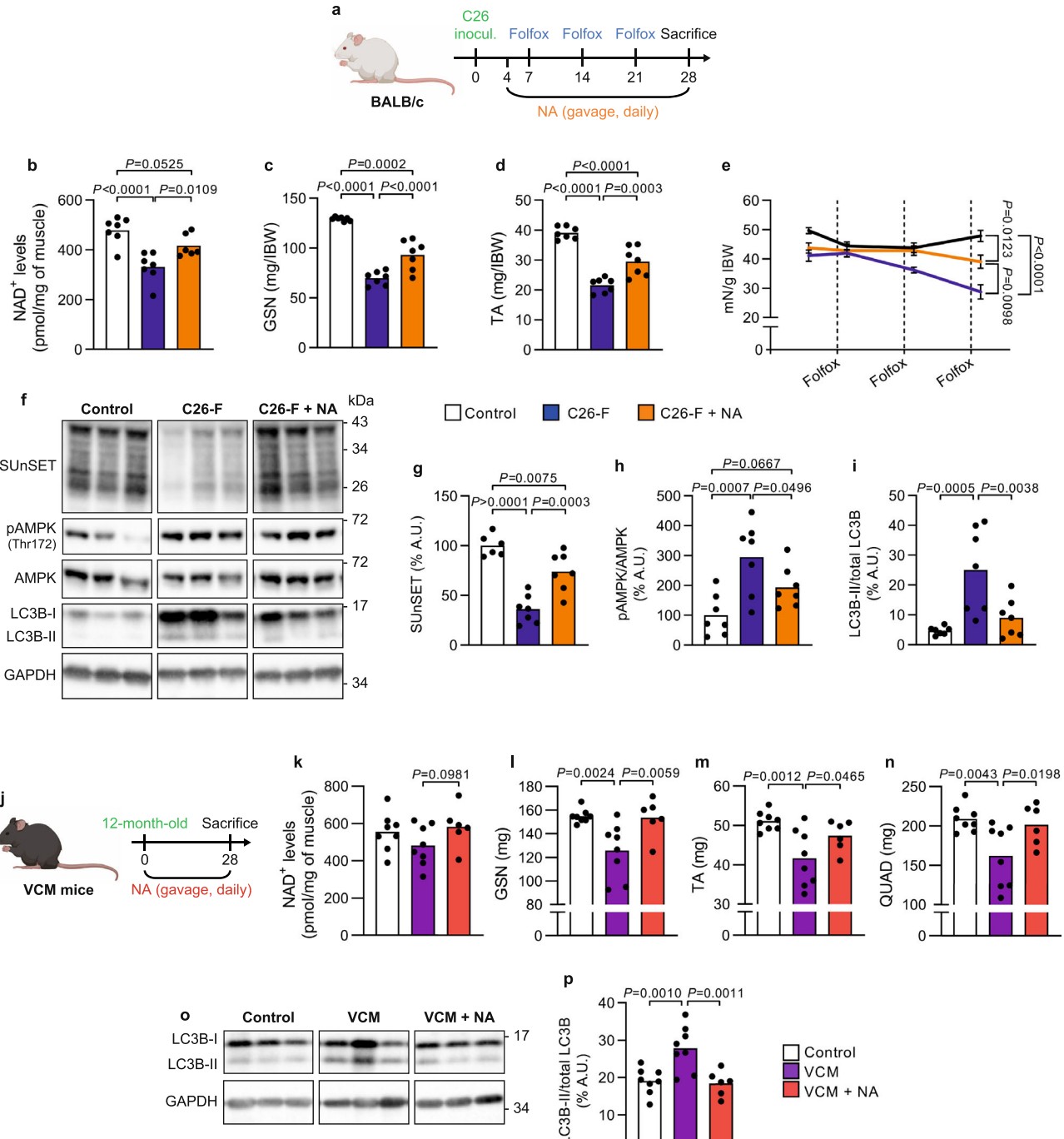

**Fig. 3 | Effects of NA treatment on muscle mass, muscle function and protein metabolism in CC mouse models. a** Study design of the C26-F model and NA treatment with three experimental groups: control, C26-F and C26-F + NA. **b** Levels of NAD⁺ represented as pmol per mg of muscle (control $n = 7$, C26-F $n = 7$, C26-F + NA $n = 6$). **c, d** *Gastrocnemius* (GSN) and *tibialis anterior* (TA) muscle wet weight normalized to initial body weight (IBW) ($n = 7$ per group). **e** Grasping strength at the start of NA treatment and the day after every Folfox administration ($n = 7$ per group). **f–i** Representative western blotting bands (**f**) and densitometry analysis of puromycin incorporation (SUnSET analysis) (**g**; control $n = 6$, C26-F $n = 7$, C26-F + NA $n = 7$), LC3B-II normalized to total LC3B (**h**; $n = 7$ per group) and p-AMPK$^{Thr172}$

normalized to total AMPK (**i**; $n = 7$ per group) protein levels. GAPDH was used as loading control. **j** Study design of the VCM model and NA treatment with three experimental groups: control, VCM and VCM + NA. **k** Levels of NAD⁺ represented as pmol per mg of muscle. **l–n** GSN, TA and *quadriceps femoris* (QUAD) muscle wet weight. **o, p** Representative western blotting bands (**o**) and densitometry analysis of LC3B-II normalized to total LC3B (**p**; control $n = 8$, VCM $n = 8$, VCM + NA $n = 6$) protein levels. GAPDH was used as loading control. Data are shown in (**b–d, g–i, k–n** and **p**) as means with individual values and in (**e**) as means ± SEM. Statistical analysis was performed using one-way ANOVA + Fisher's LSD test. Original raw data are provided as a Source Data file. NA niacin, A.U. arbitrary units.

independently of CC status, i.e., body weight loss and/or sarcopenia, this emphasizes that NAD⁺ metabolism could be a feasible target for early interventions to improve cancer patient health before overt or refractory CC ensue. In addition, our results highlight the inability of the current CC assessment procedures to detect muscle or systemic

metabolic abnormalities that strongly impair cancer patient outcome and quality of life.

A previous mouse study demonstrated that NR conversion to NAD⁺ via NRKs is active in healthy muscle and NRKs regulate metabolic adaptations upon muscle regeneration[24]. However, *Nrk2* plays a

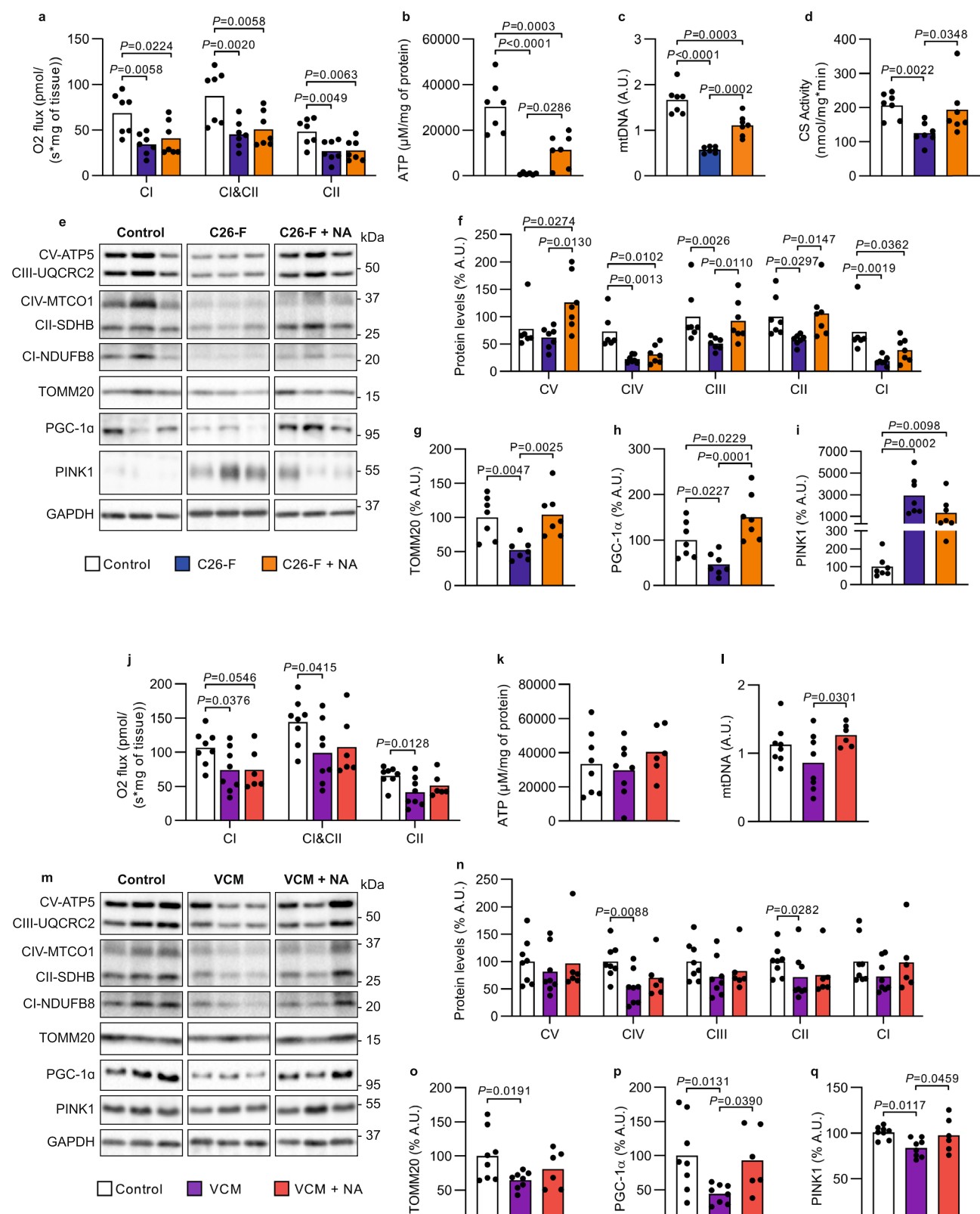

redundant role in basal muscle NAD$^+$ biosynthesis[24–26]. Interestingly, *Nrk2* is typically upregulated during metabolic energy stress and NAD$^+$ deficiency to support NAD$^+$ production[26–28]. This is contrary to our findings of consistent *Nrk2* downregulation in CC, suggesting that either the skeletal muscle has impaired adaptation to NAD$^+$ deficiency or that *Nrk2* loss plays a primary role in determining the altered NAD$^+$ and energy metabolism in CC. In line with the latter notion, low muscle

*NRK2* expression was associated with metabolite alterations in both skeletal muscle and serum of cancer patients, highlighting the future possibility to diagnose energy metabolism disturbances in CC patients with a simple venous blood sample. Overall, our results indicate that muscle NRK2 loss is a common feature of murine and human CC and that NRK2 might have a disease-specific role in the regulation of energy homeostasis.

**Fig. 4 | Impact of NA on skeletal muscle mitochondrial respiratory capacity and markers of mitochondrial biogenesis in CC mouse models. a–i** Assays run on *gastrocnemius* (GSN) muscle from control, C26-F and C26-F + NA groups ($n = 7$ per group unless differently stated): **a** mitochondrial respiration as $O_2$ flux normalized by mg of tissue; **b** ATP concentration (µM) normalized to mg of protein (C26-F $n = 6$); **c** mtDNA amount presented as a ratio of mtDNA genome to nuclear DNA genome; **d** Citrate synthase (CS) activity measured by substrate consumption over time (nmol/min) normalized by the total amount of protein (mg) in the lysate; **e** representative western blotting bands for OXPHOS complex subunits (ATP5, UQCRC2, MTCO1, SDHB and NDUF88), TOMM20, PGC-1α, PINK1 and GAPDH; **f–i** Protein content of OXPHOS complex subunits (**f**), TOMM20 (**g**), PGC-1α (**h**) and PINK1 (**i**) assessed by densitometry analysis of western blotting bands (**e**). **j–q** Assays run on GSN muscle from control, VCM and VCM + NA groups (control $n = 8$, VCM $n = 8$, VCM + NA $n = 6$): **j** mitochondrial respiration as $O_2$ flux normalized by the total mg of tissue; **k** ATP concentration (µM) normalized to mg of protein; **l** mtDNA amount presented as a ratio of mtDNA genome to nuclear DNA genome; **m** representative western blotting bands for OXPHOS complex subunits (ATP5, UQCRC2, MTCO1, SDHB and NDUF88), TOMM20, PGC-1α, PINK1 and GAPDH; **n–q** Protein content of OXPHOS complex subunits (**n**), TOMM20 (**o**), PGC-1α (**p**) and PINK1 (**q**) assessed by densitometry analysis of western blotting bands (**m**). GAPDH was used as loading control in (**f–i**) and (**n–q**). Data are shown as means with individual values. Statistical analysis was performed with one-way ANOVA + Fisher's LSD for normally distributed data and with two-tailed Kruskal–Wallis + Uncorrected Dunn's test for non-normal data. Original raw data are provided as a Source Data file. NA niacin, A.U. arbitrary units.

Beneficial effects of increasing intracellular $NAD^+$ levels have been demonstrated in various muscular and metabolic diseases[14,29–32]. In agreement with these previous studies, NA partially rescued the depleted $NAD^+$ metabolites in the skeletal muscle, counteracted muscle wasting and improved muscle function and protein synthesis in C26-F mice. Although the VCM mice showed no depletion of muscle NAD metabolites at baseline, NA restored muscle mass and normalized autophagic markers. From a speculative perspective requiring experimental demonstration, the improvements may originate, especially in C26-F mice, from restoration of mitochondrial metabolism and increased OXPHOS rather than NA having a direct effect on protein metabolism. Consequently, the energized muscles may have a lesser need for muscle degradation to fulfill systemic energy demands. Interestingly, ATP levels were dramatically reduced while PINK1 was massively increased in the C26-F skeletal muscles. These findings indicate that mitochondria from C26-F muscles may be depolarized, possibly causing ATP synthase to work in reverse mode to maintain membrane potential. Under these conditions, PINK1 is stabilized at the mitochondrial surface[33]. Our results imply that in addition to slightly improved mitochondrial function, as suggested by mildly increased CII enzymatic activity, NA treatment may also partially restore mitochondrial membrane potential, leading to normal activity of ATP synthase and the partial restoration of ATP level, even if PINK1 reduction upon NA treatment in C26-F mice did not reach a statistical significance. NA-mediated increase in mitochondrial biogenesis likely occurred via PGC-1α in both C26-F and VCM mice, similarly to the previously reported $NAD^+$ boosting with NR in rodent models[34,35]. In a recent human study from our group, NA also increased muscle mitochondrial biogenesis and improved muscle strength in healthy individuals and patients with mitochondrial myopathy[14]. Collectively, our murine study indicates that NA has a therapeutic potency on CC regardless of the muscle $NAD^+$ content, that may vary according to the severity of CC and/or the exposure to chemotherapy.

Considering that the liver performs a wide range of energetically demanding processes, it has been suggested that hepatic metabolism requires proper $NAD^+$ homeostasis[36–38]. In line with this, dysregulated mitochondrial, lipid and glucose metabolism are associated with hepatic $NAD^+$ deficiency[39,40]. Here we provide the first evidence that both severe and mild CC mouse models exhibit pronounced hepatic NAD metabolite depletion. The underlying cause for hepatic $NAD^+$ deficiency may be related to perturbed $NAD^+$ biosynthesis, at least in C26-F mice. These findings revealed that $NAD^+$ metabolism aberrations in CC are rather of systemic nature than specifically distinctive for skeletal muscle. Despite NA effectively rescued hepatic NAD metabolite levels and muscle wasting in both models, it only partially restored liver mitochondrial metabolism. Overall, a causal relationship cannot be established between liver dysfunction and muscle wasting in the currently adopted murine CC models. The lack of direct causality does not rule out the possibility of NA to improve the overall host metabolism considering its known favorable actions on the liver[41]. As the liver is often neglected in CC studies that focus mainly on the skeletal

muscle, we aim to deeper characterize its involvement in future investigations.

In conclusion, our findings encourage investigating NRK2-targeted therapeutic options to improve disturbed energy metabolism in CC. In addition, the results demonstrate that NA has a therapeutic effect on both cancer- and chemotherapy-induced cachexia in mice. The effectiveness of NA in variable experimental conditions that reflect the broad spectrum of human CC increases the translational value of our findings. Although deeper insight on $NAD^+$ metabolism impairments in human CC are still required, our study highlights the necessity of $NAD^+$ to support energy metabolism in CC and paves the way for the development of vitamin B3-based therapies to effectively target the multifaceted aspects of cancer cachexia.

## Materials and methods

All the reagents used in this work were obtained from Merck Sigma-Aldrich (St. Louis, MO, USA) unless differently indicated.

### Animals and experimental design

Experimental animals were cared for in compliance with the Italian Ministry of Health Guidelines and the Policy on Humane Care and Use of Laboratory Animals (NRC, 2011). The experimental protocols were approved by the Bioethical Committee of the University of Torino (Torino, Italy) and the Italian Ministry of Health (authentication number 579/2018-PR). The animals were maintained on a regular dark-light cycle of 12:12 h with controlled temperature (20–23 °C), humidity (40–60%) and free access to food (2018 Teklad global 18% protein rodent chow) and water during the whole experimental period. B6.Cg-Tg(Vil1-cre) 997Gum/J (Villin-Cre) and B6.Cg-Msh2tm.1Rak/J (Msh2loxP), mice were purchased from The Jackson Laboratory (Bar Harbor, CA, USA) and were crossed to obtain the Villin-Cre/Msh2loxP/loxP(VCM) offspring, leading to the conditional knock-out of the Msh2 gene in the enterocytes of the intestinal mucosa, accelerating the formation of intestinal adenomas/adenocarcinomas[16]. The presence of each transgenic construct was assessed through Melt Curve Analysis (RT-qPCR) using the following primers: Villin-Cre (forward: 5′-TTCTCCTCTAGGCTCGTCCA-3′ and reverse: 5′-CATGTCCATCAGGTTCTTGC-3′) and Msh2loxP (wild-type: 5′-GATGATGTGTGAAGCCTGCAT-3′, mutant: 5′-CCTCTTGAGGGGAATT-GAAGT-3′ and common: 5′-AGGTTAAAAACCAGAGCCTCAACT-3′).

C26-F model: 6-month-old wild-type BALB/c female mice weighing ~20 g (Charles River, Wilmington, MA) were divided into 3 groups ($n = 7$): healthy controls, C26-F and C26-F + NA. Females were used to avoid the fighting characteristic among male cagemates subjected to severe cachexia protocols. C26-F mice were subcutaneously inoculated with $5 \times 10^5$ Colon26 (C26) adenocarcinoma cells, obtained in 2005 from Prof M.P. Colombo (National Cancer Institute, Milano, Italy), on the back and treated with Folfox chemotherapy (6 mg/Kg oxaliplatin, 25 mg/Kg 5-fluorouracil, 90 mg/Kg leucovorin) at days 7, 14 and 21 after tumor inoculation. Mice of the C26-F + NA group were administered a daily dose of NA (150 mg/kg dissolved in tap water) by gavage. Healthy controls received an equal saline injection excluding

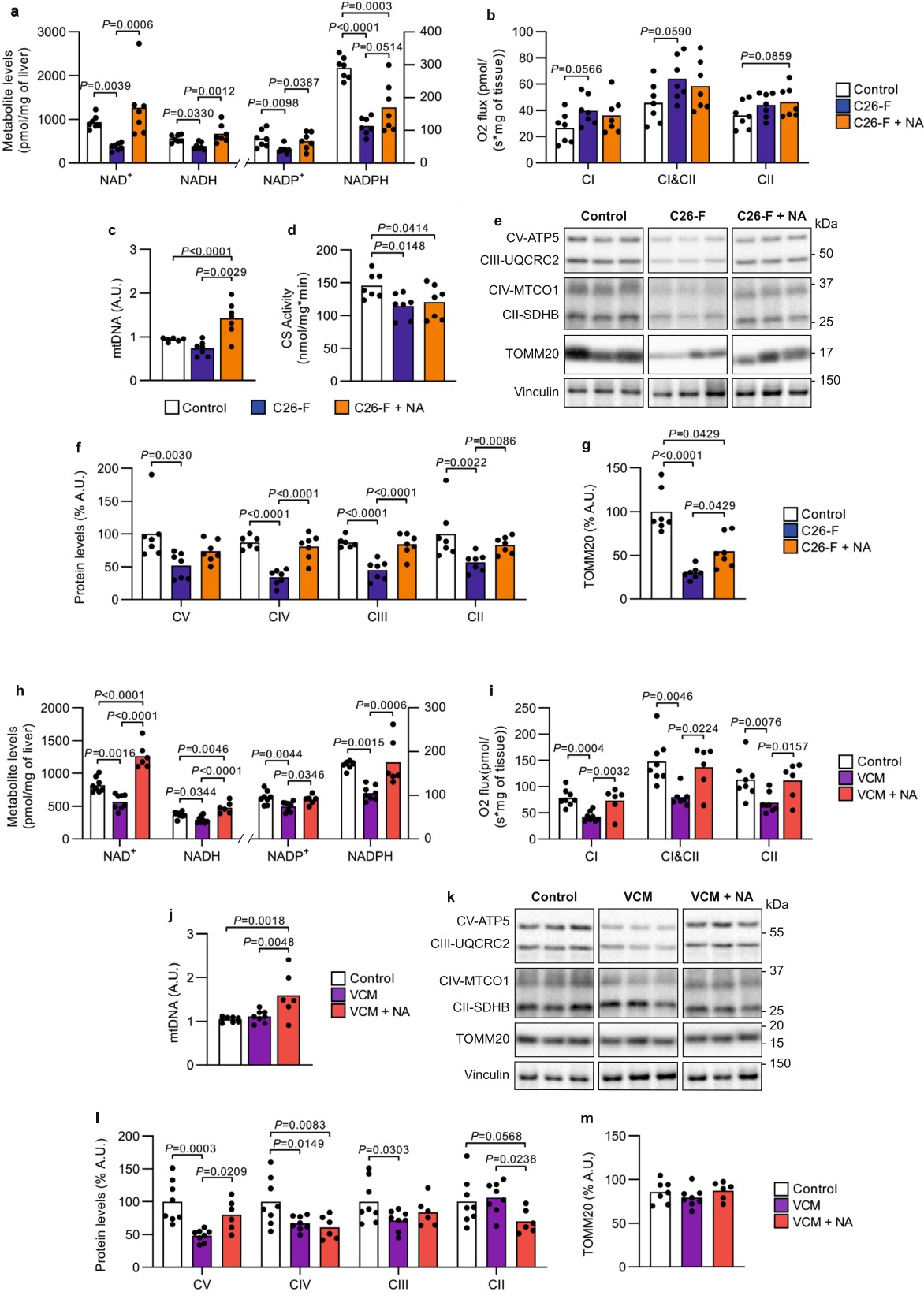

the cell inoculum and were daily treated with tap water. Grasping strength was assessed on day 0 and the day after each Folfox administration (days 8, 15 and 22). Oral treatments with NA started 4 days after tumor injection. At day 28 post-C26 implantation, mice were injected with an intraperitoneal dose of 40 μmol/Kg puromycin 30 min prior to euthanasia in order to assess the relative rate of protein

synthesis (SUnSET methodology)[42]. The amount of puromycin incorporated into nascent peptides was detected by western blotting, using a specific anti-puromycin antibody. In order to assess survival rates, 3-months-old BALB/c female mice animals were inoculated with $5 \times 10^5$ C26 cells and treated with Folfox at days 7, 14, 21 and 28 after tumor inoculation. Mice in the C26-F + NA group ($n = 11$) were daily treated

**Fig. 5 | Impact of NA on hepatic NAD+ and mitochondrial metabolism in CC mouse models. a–g** Assays run on frozen liver from control, C26-F and C26-F + NA groups ($n = 7$ per group unless differently stated): **a** levels of NAD metabolites represented as pmol per mg of tissue; **b** mitochondrial respiration as $O_2$ flux normalized by mg of tissue; **c** mtDNA amount presented as a ratio of mtDNA genome to nuclear DNA genome (C26-F $n = 6$); **d** citrate synthase (CS) activity measured by substrate consumption over time (nmol/min) normalized by mg of total protein in the lysate; **e** representative western blotting bands of OXPHOS complex subunits (ATP5, UQCRC2, MTCO1, SDHB and NDUF88), TOMM20 and Vinculin; **f–g** protein content of TOMM20 (**g**) and OXPHOS complex subunits (**f**; control $n = 6$ in UBCRC2 and MTCO1) assessed by densitometry analysis of western blotting bands (**e**). **h–m** Assays run on frozen liver from control, VCM and VCM + NA groups (control $n = 8$, VCM $n = 8$, VCM + NA $n = 6$ unless differently stated): **h** levels of NAD metabolites

represented as pmol per mg of tissue (control NADH $n = 7$ and control NADPH $n = 6$); **i** mitochondrial respiration as $O_2$ flux normalized by mg of tissue (VCM CI&CII $n = 7$); **j** mtDNA amount presented as a ratio of mtDNA genome to nuclear DNA genome; **k** representative western blotting bands of OXPHOS complex subunits (ATP5, UQCRC2, MTCO1, SDHB and NDUF88), TOMM20 and Vinculin; **l–m** protein content of TOMM20 (**m**) and OXPHOS complex subunits (**l**) assessed by densitometry analysis of western blotting bands (**k**). Vinculin was used as a loading control in (**f–g**) and (**l–m**). Data are shown as means with individual values. Statistical analysis was performed with one-way ANOVA + Fisher's LSD for normally distributed data and with two-tailed Kruskal–Wallis + Uncorrected Dunn's test for non-normal data. Original raw data are provided as a Source Data file. NA niacin, A.U. arbitrary units.

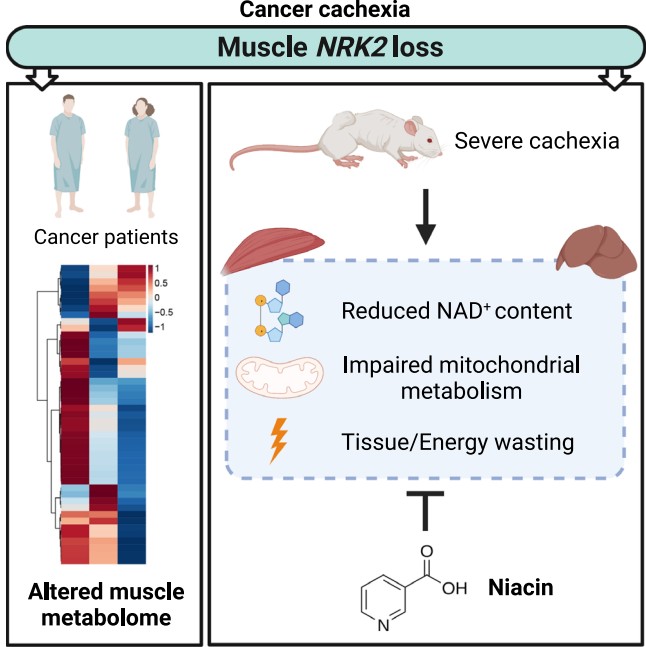

**Fig. 6 | Graphical representation of the main findings of this study.** Observations in cancer patients confirm the data obtained in three distinct murine models of experimental cancer cachexia and are supported by positive results in the intervention study involving severe (C26-F) and chronic (VCM) cancer cachexia. The beneficial effects produced by niacin on NAD+ content, mitochondrial homeostasis and energy metabolism are generalized despite model specific alterations are present.

with NA (150 mg/kg) by gavage starting 4 days after tumor injection, whereas animals in the C26-F group ($n = 12$) received water. Mice were sacrificed when a single human endpoint was reached. Endpoint criteria included pronounced body mass loss (>30% of IBW) and overall condition of the mouse (lack of grooming, kyphosis, inactivity, reduced reactivity and anorexia). At day 49, the surviving animals were euthanized.

VCM model: 12-month-old male VCM mice ($n = 6$) were treated with a daily dose of NA (150 mg/kg, up to 4.5 mg/day) by gavage for 28 days. Age and gender-matched VCM mice ($n = 8$) and Msh2^loxP/loxP mice ($n = 8$) were used as non-treated tumor-bearers and healthy controls, respectively.

KPC model: muscle samples were obtained from a previous animal experiment[15]. Briefly, 8-week-old wild-type C57BL/6 male mice were subcutaneously inoculated with $0.7 \times 10^6$ cells KPC cells ($n = 6$) derived from the primary culture of a pancreatic tumor (K-ras^LSL.G12D/+; p53^R172H/+; Pdx-Cre), and tissues were collected 5 weeks after tumor implantation. Healthy age-matched C57BL/6 mice were used as controls ($n = 6$).

In all experimental protocols, body weight and food intake were monitored every other day, and the animals were daily examined for signs of distress. At the endpoint, the mice were anesthetized with 2% isoflurane in O2, blood was collected by cardiac puncture and euthanasia was performed by cervical dislocation. Several tissues were excised, weighed, frozen in liquid nitrogen and stored at −80 °C for further analyses.

### Collection of human skeletal muscle and serum samples
The human samples originate, with some additions, from a previous study[17]. From 2015 to 2020 we enrolled consecutive patients with colorectal or pancreatic cancer and control patients undergoing surgery for benign diseases at the 3rd Surgical Clinic of the University Hospital of Padova. Sex was self-reported; the results of the study were analyzed in an aggregated form in order to obtain a sufficient numerosity for statistical significance of the results. Sex-based differences will be assessed in a follow-up study with a larger patient cohort. The research project was approved by the Ethical Committee for Clinical Experimentation of Padova (protocol number 3674/AO/15). All patients voluntarily joined the protocol without any compensation and according to the guidelines of the Declaration of Helsinki and the written informed consent was obtained from participants. The muscle biopsy was performed at the time of the planned surgery by a cold section of a *rectus abdominis* fragment ($1 \times 0.5$ cm) immediately frozen and conserved in liquid nitrogen for gene expression analysis and metabolome profiling. Serum samples were obtained from blood samples retrieved prior to any surgical manipulation. Demographics and clinical data, including medications and comorbidities noted as having potential confounding effects on skeletal muscle homeostasis[17] were collected from all patients (Supplementary Table 1). Cancer patients were classified as cachectic in cases of >5% weight loss in the 6 months preceding surgery, >2% weight loss with either body mass index (BMI) < 20 or low muscle mass defined by the skeletal muscle index (SMI) cut-offs described by Martin et al.[43]. SMI values were quantified using the preoperative CT scans as previously described[17]. Based on gene expression analysis, we selected 10 cancer patients with the highest (comparable to healthy controls) and 10 with the lowest (almost tenfold decrease) *NRK2* levels (Supplementary Table 2). We performed metabolome profiling of muscle and serum samples in these two subgroups.

### Metabolome analysis
About 10 mg of skeletal muscle or 50 μl of serum were used for metabolite extraction and analysis. The samples were flash frozen upon collection and sent for further processing to the Metabolomics Expertize Center, VIB Center for Cancer Biology, KULeuven Department of Oncology, Leuven, Belgium. The extraction was performed adding 99 or 19 volumes (for muscles or sera, respectively) of 80% methanol, containing 2 uM d27 myristic acid as internal standard. The mixture was centrifuged at 20,000 × $g$ for 15 min at 4 °C to precipitate proteins and insoluble material, the supernatant transferred to a fresh

new tube. 10 µl of each sample was loaded into a Dionex UltiMate 3000 LC System (Thermo Scientific Bremen, Germany) equipped with a C-18 column (Acquity UPLC -HSS T3 1. 8 µm; 2.1 × 150 mm, Waters) coupled to a Q Exactive Orbitrap mass spectrometer (Thermo Scientific) operating in negative ion mode. A step gradient was carried out using solvent A (10 mM TBA and 15 mM acetic acid) and solvent B (100% methanol). The gradient started with 5% of solvent B and 95% solvent A and remained at 5% B until 2 min post injection. A linear gradient to 37% B was carried out until 7 min and increased to 41% until 14 min. Between 14 and 26 min the gradient increased to 95% of B and remained at 95% B for 4 min. At 30 min the gradient returned to 5% B. The chromatography was stopped at 40 min. The flow was kept constant at 0.25 mL/min at the column was placed at 40 °C throughout the analysis. The MS operated in full scan mode (m/z range: [70.0000–1050.0000]) using a spray voltage of 4.80 kV, capillary temperature of 300 °C, sheath gas at 40.0, auxiliary gas at 10.0. The AGC target was set at 3.0E + 006 using a resolution of 140000, with a maximum IT fill time of 512 ms. Data collection was performed using the Xcalibur software (Thermo Scientific). The data were obtained by integrating the peak areas (El-Maven – Polly - Elucidata). Data analysis was performed using the free online resource https://www.metaboanalyst.ca/ version 5.0.

## Assessment of NAD metabolite levels

$NAD^+$, NADH and the phosphorylated metabolites $NADP^+$ and NADPH were measured from pulverized *gastrocnemius* (GSN) muscle and liver with a slightly modified conventional colorimetric method[4] (for further information, see https://www.nadmed.fi/). NAD metabolite levels were normalized to tissue mass used for analysis or total protein content of the sample. Protein concentration was determined according to the Bradford method using the Bio-Rad reagent.

## RNA isolation and RT-qPCR analysis

Approximately 30 mg of GSN muscle and liver were lysed and processed to isolate high-quality RNA using the standard phenol-chloroform method. The RNA concentration was quantified by means of spectrophotometry. Total RNA was retro-transcribed using a cDNA synthesis kit (Bio-Rad or Qiagen, Hilden, Germany) and transcript levels were determined by RT-qPCR using the SsoAdvanced SYBR Green Supermix (Bio-Rad) or Maxima SYBR Green qPCR Master Mix (Thermo fisher Scientific) and the CFX Connect Real-Time PCR Detection System (Bio-Rad) with 10 ng of cDNA per well. Every RT-qPCR was validated by analyzing the respective melting curve and run in parallel to no reverse transcriptase control (NRT) to exclude potential artifacts from genomic DNA contamination. Gene expression was normalized to the geometric mean of housekeeping gene expression and represented as relative expression according to primer efficiency assessed using serial dilutions of pooled samples (standard curve method). Data analysis was conducted in Microsoft Excel and qBASE+ software (Biogazelle). As for human muscle biopsies, total RNA was extracted from ~20 mg of *rectus abdominis* muscle using TRIzol (Thermo Fisher Scientific). 1 ug of RNA was reverse transcribed using the SuperScript IV Reverse Transcriptase (Thermo Fisher Scientific). Gene expression was analyzed by qRT-PCR using the PowerUp SYBR Green Master Mix (Applied Biosystems). Data were normalized to *ACTB* gene expression and displayed as fold change over control (healthy) group. Primer sequences are listed in Supplementary Table 3.

## PARP activity

PARP activity was analyzed from pulverized liver and GSN muscle utilizing HT Colorimetric PARP/Apoptosis Assay Kit (R&D Systems, Minneapolis, MN, USA) according to manufacturer's instructions ($n = 6$–8 per group). Data were normalized to protein concentration of the samples quantified according to the Bradford method using the Bio-Rad reagent.

## Mitochondrial respiration of frozen tissue

CI, CI&CII and CII mediated mitochondrial respiration was measured from frozen GSN muscle and liver samples of C26-Folfox and VCM models with Oroboros O2k High-Resolution Respirometry (Oroboros instruments, Innsbruck, Austria) by using Respirometry in Frozen samples (RIFS) -protocol by Acin-Perez et al.[44] modified for the Oroboros apparatus. This frozen RIFS protocol provides information about mitochondrial respiration uncoupled from ATP synthesis. During the measurements, the Oroboros chamber was filled with 2 ml of mitochondrial respiration medium MiR-05 (0.5 mM EGTA, 3 mM $MgCl_2*6H_2O$, 60 mM lactobionic acid, 20 mM taurine, 10 mM $KH_2PO_4$, 20 mM HEPES, 110 mM D-sucrose and 1 g/l of bovine serum albumin, pH = 7.1) and the runs were carried out at 37 °C under constant stirring (750 rpm) with oxygen concentration above 100 µM. Pulverized frozen tissue samples were dissolved in MiR-05 buffer immediately after taken out from the freezer with a ratio of 1 mg sample to 100 ul buffer. After mixing, the sample lysates were allowed to dissolve at +4 °C for 1 h before respirometry measurement. The amount of lysate used in the measurement was 100 ul for skeletal muscle and 50 ul for liver, corresponding 1 mg and 0.5 mg of tissue, respectively. Oxygen flux was calculated as means over ~1 min within each measured parameter after a stable flux was achieved. Residual oxygen flux was subtracted from all values. Oxygen flux was quantified using the DatLab analysis software and data were normalized per mg of tissue. Respirometry measurements were conducted from one or two replicates per sample and in case of the latter one, the mean of the two values were chosen.

## ATP quantification

Intracellular ATP concentration was measured from GSN muscle with a commercial bioluminescence kit (ATP Bioluminescence Assay Kit CLS II; Roche, Basel, Switzerland) according to the manufacturer's instructions with slight modifications. In short, 10 mg of powderized muscle was dissolved in phosphate buffered saline (PBS; 10% wt/vol). Samples were then incubated on ice for 30 min with agitation. Homogenates were then diluted with 9 volumes of boiling 100 mM Tris, 4 mM EDTA (pH 7.75) and incubated for 2 min at 100 °C. Differing from the manufacturer's instructions, supernatants were then collected twice after two centrifugations of 1 min at 1500 × g to yield clear supernatant. Finally, 50 µl of the supernatant and 50 µl of luciferase reagent were loaded on a white 96-well plate (Corning, NY, USA). Luminescence was measured with an integration time of 3 s at 562 nm with Enspire plate reader. ATP concentrations were obtained from a log-log plot of the standard curve and normalized to protein concentration of the samples quantified according to the Bradford method using the Bio-Rad reagent.

## Mitochondrial DNA amount quantification

Total DNA, including mitochondrial DNA, was extracted from ~10 mg of pulverized GSN muscle and ~3 mg of pulverized liver from C26-F and VCM mice with the standard phenol-chloroform method followed by ethanol precipitation. The amount of mtDNA was determined as the ratio of mitochondrial rRNA 16 s and cytochrome c oxidase subunit II (Cox2) genomic regions, to the geometric mean of nuclear uncoupling protein 2 (Ucp2) and hexokinase-2 (Hk2) genomic regions using RT-qPCR. Primer sequences are listed in Supplementary Table 3. RT-qPCR was carried out in triplicates with 2 ng of template DNA per well using Maxima SYBR Green qPCR Master Mix (Thermo fisher Scientific). Data analysis was conducted with standard curve method with qBASE+ software (Biogazelle).

## Biochemical analysis of citrate synthase (CS) and mitochondrial respiratory chain complexes

Approximately 30 mg of either GSN muscle or liver were homogenized using an Elvehjem potter in ice-cold 320 mM Sucrose, 1 mM EDTA,

10 mM Tris-HCl, pH 7.4 buffer. Samples were centrifuged at $800 \times g$ for 5 min, the supernatant was collected and protein concentration was quantified using the BCA assay. The homogenate was used to measure CS activity as previously described[45]. The lysates were then subjected to three freeze-thaw cycles and the activity of single mitochondrial complexes was assessed spectrophotometrically as previously reported[45].

## Western blotting

Approximately 50 mg of GSN muscle and liver were mechanically homogenized using bead homogenizer in RIPA buffer (50 mM Tris-HCl pH 8.0, 5 mM EDTA pH 8.0, 1% Igepal CA-630, 0.5% sodium deoxycholate, 0.1% SDS) containing protease inhibitors (0.5 mM PMSF, 0.5 mM DTT, 2 µg/ml leupeptin, 2 µg/ml aprotinin) and phosphatase inhibitors (P0044). Next, homogenates were sonicated for 10 s at low intensity, centrifuged at 15,000 g for 5 min at 4 °C and the supernatant was collected. Total protein concentration was quantified with Bradford reagent (Bio-Rad) using BSA as protein concentration standard. Equal amounts of protein (10–30 µg) were heat-denatured (except when assessing OXPHOS expression) in sample-loading buffer (50 mM Tris-HCl, pH 6.8, 100 mM DTT, 2% SDS, 0.1% bromophenol blue, 10% glycerol), resolved by SDS-PAGE electrophoresis (4561086, Bio-Rad) and transferred to nitrocellulose membranes (1704159, Bio-Rad). Membranes were blocked with 5% nonfat dry milk in Tris-buffered saline containing 0.05% Tween (TBS-Tween) and then incubated overnight with antibodies directed against specific proteins: AMPK (1:1000, 07-181, Millipore, polyclonal), p-AMPK$^{Thr172}$ (1:1000, #2535, Cell Signaling, clone 40H9), GAPDH (1:10000, G8795, clone GAPDH-71.1), LC3B (1:1000, L7543, polyclonal), NRK2 (1:1000, produced in Dr. Gareth G Lavery's lab, polyclonal), OXPHOS Antibody Cocktail (1:1000, ab110413, Abcam), PGC-1α (1:1000, AB3242, Merck Millipore, polyclonal), PINK1 (1:500, SAB2500794, polyclonal), Puromycin (1:1000, EQ0001, Kerafast, clone 3RH11), TOMM20 (1:1000, ab186735, Abcam, clone EPR15581-54) and Vinculin (1:2000, sc73614, Santa Cruz Biotechnology, clone 7F9). Goat anti-mouse, goat anti-rabbit and rabbit anti-goat HRP-conjugated IgGs (1:8000, Bio-Rad) were used as secondary antibodies, except for OXPHOS Antibody Cocktail which was analyzed using Mouse TrueBlot ULTRA: Anti-Mouse Ig HRP (1:1000, 18-8817-30, Kerafast, clone eB144). Three 5 min washes with TBS-Tween were performed after each antibody incubation. After incubation with Clarity Western ECL substrate (170-5061, Bio-Rad), bands were developed using the ChemiDoc XRS + imaging system (Bio-Rad). Densitometric analysis on the obtained images was performed using the Image Lab software (Bio-Rad). Uncropped and unprocessed gels are included in the Source Data file.

## Liver glycogen and glutathione content

Liver glycogen concentration was assessed using a commercially available system (MAK016 Glycogen assay Kit). Briefly, liver fragments of about 50 mg were cold homogenized in water (10% w/vol) with a bead homogenizer (Bullet Blender, New Advance, Troy, NY, USA), boiled for 5 min and centrifuged for 5 min at $13,000 \times g$. The supernatant was collected and diluted 100-fold before adding 10 µl to a 96-well plate. The assay was performed following manufacturer's instructions and using a glycogen titration curve in order to extrapolate quantitative data.

Glutathione was determined as previously described[46], with slight modifications[47]. Briefly, liver fragments of about 50 mg were cold homogenized in water (10% w/vol) with a bead homogenizer, deproteinized on ice using 5% metaphosphoric acid and centrifuged at $15,000 \times g$ for 2 min at 4 °C. The supernatants were treated with 4 M triethanolamine to reach pH 7.4. GSH concentration was determined after 2 min incubation with 5,50-dithiobis-2-nitrobenzoic acid (DTNB) by measuring the production of 50-thio-2-nitrobenzoic acid (TNB) at 412 nm on a 96-well microplate reader. Suitable volumes of diluted glutathione reductase (6 U/mL) and of NADPH (4 mg/mL) were then added to evaluate total glutathione level (GSH + GSSG). GSSG content was calculated by subtracting GSH content from total glutathione levels.

## Data representation and statistics

Data are presented using bar (mean) and dot plots (individual values) unless differently stated. Data representation and statistical tests were performed with Prism (version 9, GraphPad) software. Outliers were identified using ROUT (Q = 1%) and excluded from the analysis. The normality of distributions was evaluated by the Shapiro–Wilk test. Unless otherwise stated in the figure legend, the significance of the differences was evaluated by appropriate two-tailed statistical tests: Student's "$t$" test or analysis of variance (ANOVA) for normal distribution and Mann–Whitney test or Kruskal–Wallis test for non-normal distribution. ANOVA was followed by the Fisher's Least Significant Difference (LSD) test, whereas the Kruskal–Wallis test was followed by the Uncorrected Dunn's test to assess differences of planned comparisons among groups.

## Reporting summary

Further information on research design is available in the Nature Portfolio Reporting Summary linked to this article.

## Data availability

All data are included in this paper and Supplementary Information. Source Data underlying figures are provided as a Source Data file. Metabolome raw data are included in the Supplementary Data 1 file. Source data are provided with this paper.

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

## Acknowledgements

We are grateful to Dr. Gareth G Lavery (Department of Biosciences, Nottingham Trent University) who kindly donated the antibody against the NRK2 protein. Also, we thank Valentina Audrito (DISIT, University of Piemonte Orientale) for reviewing and advising on the first draft of the paper. This work was supported by Fondazione AIRC (IG 2018—ID. 21963 project, PI: F.P.), the Finnish Cancer Foundation, Finnish Cancer Center FICAN South (PIs: E.P. and Dr. Tommi Järvinen, respectively), the Academy of Finland (profi6 336449 to E.P.), The Finnish Medical Foundation (doctoral research grant to N.P.), and by two post-doctoral Fellowships from Fondazione Umberto Veronesi (ID2496 and ID3519 to R.S). Figures 3a, j and 6 were created with BioRender.com.

## Author contributions

M.B., N.P., R.S., E.P., and F.P. designed the experiments. M.B., N.P., C.F., K.T., M.Y.H., S.C., and R.S. performed experiments and data analyses, M.B., C.F., and F.P. prepared and carried out the animal experiments. M.B., N.P., E.P., and F.P. wrote the paper. S.Z., L.M., P.E.P., R.K., C.V., M.S., and R.S. provided material and technical support, P.E.P., R.K., C.V., M.S., J.J.H., and R.S. provided advice and comments. E.P. and F.P. conceived the ideas and provided funding. R.S., E.P., and F.P. equally contributed and jointly supervised this work.

## Competing interests

The authors declare no competing interests.
