## [Peer Review File · Nature Communications]

NAD⁺ repletion with niacin counteracts cancer cachexiaEditorial Note: Parts of this Peer Review File have been redacted as indicated to maintain confidentiality.

REVIEWER COMMENTS

Reviewer #1 (Remarks to the Author); expert in metabolism and metabolomics:

In this manuscript, Beltrà et al reported that depletion of NAD⁺ and downregulation of Nr2 are common features of different mouse models and cachectic cancer patients. They further found that NRK2 loss is associated with a unique signature of patient muscle metabolism, indicating muscle energy failure and protein hypercatabolism. Subsequently, they show that Niacin administration can rescue skeletal muscle NAD⁺ levels, ameliorate muscle wasting and improve protein metabolism in experimental cancer cachexia. Finally, they demonstrated that Niacin improves skeletal muscle mitochondrial biogenesis and partially improves hepatic mitochondrial alterations in experimental cancer cachexia. This study shows interesting observations of altered muscle metabolism in cancer cachexia. However, the lack of mechanistic studies of this article limits its publication in Nature Communication. I have the following concerns and suggestions:

1. In Fig. 1, please provide the correlated body weight when muscle NAD⁺ level (Fig. 1a) and Nr2 expression level (Fig. 1b) were measured in different tumor models. For the mouse models, both C26-F and KPC are allografts. It will be interesting to know if same phenomenon can be observed in KPC GEMM. In addition, for the VCM mice, what is the time point of NAD⁺ measured in Fig. 1a? Is NAD⁺ level dropped when CC occurs in VCM mice at late stage of tumor progression?
2. In Fig. 2, samples from human patient show the changes of glycolic and TCA cycle intermediates between low NRK2, healthy, and high NRK2 patients. Can same metabolic change be observed in muscle and serum from mouse models of C26-F and KPC?
3. The expression of enzymes associated with skeletal muscle mitochondrial biogenesis was reduced in C26-F mouse model, which was rescued by Niacin treatment (Fig 4). However, how does this alteration affect muscle mitochondrial metabolism? This can be examined by in vivo isotope tracing and metabolic flux analysis. Alternatively, muscle oxygen consumption rate should be examined. Same assay should be performed in liver tissues. If reduced muscle NAD⁺ in C26-F model is associated with impaired mitochondria as claimed by this manuscript, what is energy production or ATP level in control, C26-F and Niacin treatment groups?

Reviewer #2 (Remarks to the Author); expert in cancer-associated cachexia:

Beltra, Pöllänen, and colleagues present a manuscript describing a role for niacin (NA) supplementation in reducing experimental cancer cachexia. This work follows their previous work, which demonstrated in reduced NAD⁺ and reduced Nr2 in colon-26 (C-26) tumor-bearing mice. In this manuscript, the authors describe similar results in C-26 mice treated with folfox (C-26F), and also reduced NAD⁺ in KPC pancreatic cancer mice. The authors report that niacin (NA) supplementation reduces muscle wasting in C-26F and VCM mice, despite the fact that NAD⁺ levels are not significantly reduced in VCM mice, and suggest that NA may be sufficient to improve energy metabolism in cancer patients to prevent cachexia. They support this claim with data demonstrating

clear build-up of citrate in the TCA cycle and glycolysis products in muscle from cancer patients with low NrK2 compared to healthy individuals or cancer patients with high NrK2, indicating clear changes in metabolism associated with NrK2 gene expression.

Major concerns

Because the difference in tumor burden is not statistically significant does not mean that it is not physiologically meaningful (supplemental Figure 2h). While the error bars are large, the mean difference is perhaps 30% of the untreated tumor size, which is likely physiologically meaningful. The fact that nicotinamide riboside was unable to rescue cancer cachexia increases concerns that the protection in muscle mass is due to tumor-effects, not muscle effects, and this concern is compounded by the ability of NA to afford significant protections to VCM mice without the expected changes in NAD⁺-associated markers. Tumor-intrinsic versus tumor-extrinsic effects are important to appreciate because the likely therapeutic potential is higher with tumor-extrinsic effects. To demonstrate true independence from tumor-burden, the relationship between muscle weight and tumor burden should be assessed for linearity, similar to PMID 32407241. Demonstrating that circulating myostatin or IL-6 do not change with NA treatment would further support the idea that the effects of NA are tumor-independent. Further, if possible, differences in tumor burden should be assessed in the VCM animals.

The increase in serum ATP in low NRK2 patients is perplexing to me, and is somewhat confusing, as I would expect circulating energy levels to be similar to muscle ATP levels. Is the hypothesis that there is more circulating ATP because these patients are required to make ATP outside of their muscles, due to the TCA cycle defect? At a minimum, this would be a valuable discussion point.

The heterogeneity of patient NrK2 expression is not particularly surprising (Figure 2a), but is there an opportunity to learn from it, perhaps by associating NrK2 expression to metabolite concentrations? Similarly, there is significant heterogeneity of weight loss in the low NRK2 patients utilized in Figure 2c. Is there an opportunity to leverage this heterogeneity and link high weight loss to NAD⁺ levels?

It is intriguing that both NRK2 high and NRK2 low patient muscles demonstrated lower NAD⁺ content than muscle from healthy individuals (Figure 2f). This would make for an interesting discussion point.

Niacin seems to be somewhat less effective in the VCM mice, particularly in returning mitochondrial measures to control levels. Is the hypothesis that this is because significant weight loss has already occurred prior to 12 months of age in these mice? Only part of the cachexia is occurring during the treatment window? Because NAD⁺ actually needs to be decreased for niacin supplementation to be effective (Figure 1a)?

Minor concerns

The language surrounding the depletion of NAD⁺ in “severe” cachexia (line 92) is likely overly strong, as the VCM mice showed much lower muscle masses than their controls, but did not display reduced NAD⁺.

Additional clarity would be appreciated on the method of normalization for Nr2f1 gene expression from patient samples and the conversion to AU. Further, while the methods are clear that Nr2f1 gene expression, not protein expression, was measured in muscle from cancer patients, the results could be somewhat more clear.

Technically it is a trend towards further downregulation of Nr2f1 in cachectic patients compared to pre-cachectic patients (line 109, Figure 2a)

While I agree that Fig. 2e-h are consistent with altered energy metabolism (line 118), calling these data evidence of a hypercatabolic state, which technically indicates protein breakdown, is not supported. Similarly, hypercatabolism is not supported in line 120.

The chemotherapy status (treated or not) of patients should be added to supplementary table 1, as should the breakdown of pancreatic and colon cancer patients. Similar information should be added to supplementary table 2.

That C-26F experiments were conducted only in female mice and VCM mice were only in males should be more prominently discussed in the results section, as a sexual dimorphism cannot be excluded.

In the discussion line 320, I am not sure that the lack of data on liver contributing to muscle wasting in the cancer cachexia models used can be used to exclude that NA supplementation improves muscle wasting at least in part through a liver-dependent mechanism, particularly given the results from Figure 5. This sentence could use additional support or revision.

RESPONSE TO REVIEWERS' COMMENTS

We really appreciate the reviewer's valuable feedback and constructive suggestions aimed at improving the quality of this manuscript. In this response we have attempted to address each comment, mainly by providing the results of new experiments or the deeper analysis of pre-existing samples for better supporting our conclusions, considering every criticism and revising the text accordingly. In this letter the reviewers' comments are numbered and reported in italics and our responses follow below. Corresponding changes are reported in red in the revised manuscript text file. A clean version of the manuscript is also available.

Reviewer #1 (*Remarks to the Author*); expert in metabolism and metabolomics:

In this manuscript, Beltrà et al reported that depletion of NAD+ and downregulation of NrK2 are common features of different mouse models and cachectic cancer patients. They further found that NRK2 loss is associated with a unique signature of patient muscle metabolism, indicating muscle energy failure and protein hypercatabolism. Subsequently, they show that Niacin administration can rescue skeletal muscle NAD+ levels, ameliorate muscle wasting and improve protein metabolism in experimental cancer cachexia. Finally, they demonstrated that Niacin improves skeletal muscle mitochondrial biogenesis and partially improves hepatic mitochondrial alterations in experimental cancer cachexia. This study shows interesting observations of altered muscle metabolism in cancer cachexia. However, the lack of mechanistic studies of this article limits its publication in Nature Communication. I have the following concerns and suggestions:

We are grateful to the reviewer for the recap of the main findings and strengths of our research. In the following point-by-point reply we address each single concern, hopefully releasing the reserve on the suitability for publication.

1a. *In Fig. 1, please provide the correlated body weight when muscle NAD+ level (Fig. 1a) and NrK2 expression level (Fig. 1b) were measured in different tumor models.*

The correlated body weight and muscle mass data are now available in the manuscript in Supplementary Fig. 1, also reported below in Fig. 1. As the reviewer can observe, the body weight is not comparable among the distinct animal models, while muscle atrophy is a constant feature (Fig. 1) and has been used to define the occurrence of cachexia.

UNIVERSITA' DEGLI STUDI DI TORINO
DIPARTIMENTO DI SCIENZE CLINICHE E BIOLOGICHE

Unità di Medicina Sperimentale e Patologia Clinica – C.so Raffaello 30, 10125 Torino

Fig. 1. a Final body weight of C26-F ($n=7$), KPC ($n=6$) and VCM ($n=8$) mice expressed as percentage of the mean of their respective control ($n=7, 6, 8$) group. b Wet weight of *tibialis anterior* (TA) and *gastrocnemius* (GSN) muscles of C26-F mice ($n=7$) normalized by initial body weight (IBW) and represented as a percentage of the mean of the control ($n=7$) group. c Wet weight of GSN and *quadriceps femoris* (QUAD) muscles of KPC mice ($n=7$) normalized by IBW and represented as a percentage of the mean of the control ($n=7$) group. d Wet weight of TA, GSN and QUAD muscles represented as a percentage of the mean of the control ($n=8$) group.

1b. For the mouse models, both C26-F and KPC are allografts. It will be interesting to know if the same phenomenon can be observed in KPC GEMM.

We agree with the reviewer that modeling cancer with tumor allografts in syngeneic mice has limitations as compared to spontaneous tumors developed in GEMMs, including the pace of tumor growth and the ectopic vs orthotopic localization. However, the main reason upon the choice of the currently adopted models was to use a broad spectrum of cachexia conditions comparing on the one side a severe, rapid and well established cachexia model (C26-F) with a slow paced model of spontaneous tumor growth, allowing us to identify several molecular analogies, even if the muscle NAD⁺ levels did not significantly drop in VCM mice. On the other side, the inclusion of a model for pancreatic cancer allows a direct comparison with the human cohort analyzed in this study that includes both intestinal and pancreatic cancer patients. We tried to contact groups working with the KPC GEMM and found a potential collaborator **[REDACTED]**

UNIVERSITA' DEGLI STUDI DI TORINO
DIPARTIMENTO DI SCIENZE CLINICHE E BIOLOGICHE

Unità di Medicina Sperimentale e Patologia Clinica – C.so Raffaello 30, 10125 Torino

willing to send muscle samples from KPC mice as soon as possible, compatible with the time frame required to obtain a sufficient numerosity and a reasonable age for developing cancer and cachexia. We foresee being able to obtain such results the earliest in the third quarter of 2023. We hope the reviewer and the editor will consider such results non critical for the current manuscript. In the attempt to extend our observations to clinically relevant cancer models, preliminary data (Fig. 2, see below) on mice bearing orthotopic colon cancers obtained by injecting VPF6 cells (p53R172H BRAFV637E mutant) in immunocompetent mice confirm that muscle *Nrk2* loss is a common feature in tumor-bearing animals.

Fig. 2. Expression of *Nrk2* gene in the skeletal muscle of control (n=5) and VPF6 orthotopic tumor-bearing mice (n=5). Data are normalized to housekeeping gene expression and displayed as fold change

1c. In addition, for the VCM mice, what is the time point of NAD⁺ measured in Fig. 1a? Is NAD⁺ level dropped when CC occurs in VCM mice at late stage of tumor progression?

We thank the reviewer for the relevant question. We have provided information about the age of the mice in the figure legend (line 105). Regarding the possibility to observe NAD⁺ loss in VCM mice at late tumor progression stages, we agree with the reviewer's speculation of a direct relationship between cachexia severity and NAD⁺ loss, however, we unfortunately have no clues supporting such hypothesis. The VCM model is new and difficult to manage, requiring a long planning for obtaining homogeneous animal cohorts. At the moment we have no animals available older than 13 months. We may potentially provide such data in the future (considering again that we presented data from 13 m old mice, impossible to repeat rapidly). Moreover, the heterogeneity of the spontaneous tumors obtained and the mild cachexia status of the animals may have affected the results on NAD⁺ levels. It is noteworthy that despite the mild phenotype and the heterogeneity, muscle *Nrk2* gene expression is reduced in the skeletal muscle of VCM mice, providing a molecular hallmark of the disrupted NAD⁺ metabolism, as observed in all the other experimental models and in cancer patients.

2. In Fig. 2, samples from human patient show the changes of glycolic and TCA cycle intermediates between low NRK2, healthy, and high NRK2 patients. Can same metabolic change be observed in muscle and serum from mouse models of C26-F and KPC?

UNIVERSITA' DEGLI STUDI DI TORINO
DIPARTIMENTO DI SCIENZE CLINICHE E BIOLOGICHE

Unità di Medicina Sperimentale e Patologia Clinica – C.so Raffaello 30, 10125 Torino

We tried to compare the current data from human samples (healthy vs low *Nrk2*) with C26 and C26-F samples obtained in our laboratory (PMID: 33670497) or by collaborators (PMID: 30680954), while we did not find data from muscle and serum of KPC cachectic mice. As presented below in Table 1, few changes are consistent between mice and humans, while several are not. On the one hand, such inconsistency is not surprising, keeping in mind the acute (mice) vs chronic (humans) cancer disease and the lack of studies finding consistent metabolomic changes between mice and humans, except a recent one on type 2 diabetes (PMID: 34185398). The other way around, looking at the metabolomic signatures from a speculative standpoint and keeping in mind the chronic vs acute settings, the results in humans showing mostly accumulation of metabolites suggest a severe impairment of metabolic activity, eventually leading to muscle energy failure. In mice, the same final event looks to be the consequence of severe depletion of metabolites, likely due to the higher impact of cancer-induced anorexia and hypermetabolism. The coexistence of muscle *Nrk2* loss in mice and humans, despite the distinct metabolomic signature, suggests that muscle adaptation upon tumor growth is preserved across species and better traces muscle energy metabolism alterations than the metabolomic profile. In conclusion, metabolomics data are unlikely to predict whether a given treatment effective in mice will prove effective in humans, while such analysis is useful for the discovery of the association between molecular and biochemical changes.

	MUSCLE		SERUM	
	human	mouse	Human	Mouse
Phosphoenolpyruvate	Red	White	White	White
Fructose 6-phosphate	Red	White	White	White
Dihydroxyacetone phosphate	Red	White	White	White
Pyruvate	Red	White	Blue	Blue
Citrate	Red	White	White	Blue
Succinate	White	Blue	Red	Blue
L-Glutamate	Red	Blue	Red	White
L-Glutamine	Red	Red	Red	White
L-Phenylalanine	Blue	Red	White	White
L-Aspartate	White	White	Red	Blue
L-Alanine	Red	White	White	White
L-Leucine	Blue	Red	White	Red
L-Valine	Blue	Red	White	Blue
L-Arginine	Red	White	White	White
L-Asparagine	Red	White	White	Blue
L-Lysine	Red	Red	White	White
ADP	Red	White	White	White

Table 1. Comparison of the metabolomics analysis of human samples (healthy vs low *Nrk2*) in the present study with previously collected C26 and C26-F samples (control vs C26-F) obtained in our laboratory (PMID: 33670497) or by collaborators (PMID: 30680954). Color code: increased (red), decreased (blue), unchanged or undetectable (white).

UNIVERSITA' DEGLI STUDI DI TORINO
DIPARTIMENTO DI SCIENZE CLINICHE E BIOLOGICHE

Unità di Medicina Sperimentale e Patologia Clinica – C.so Raffaello 30, 10125 Torino

3. The expression of enzymes associated with skeletal muscle mitochondrial biogenesis was reduced in C26-F mouse model, which was rescued by Niacin treatment (Fig 4). However, how does this alteration affect muscle mitochondrial metabolism? This can be examined by in vivo isotope tracing and metabolic flux analysis. Alternatively, muscle oxygen consumption rate should be examined. Same assay should be performed in liver tissues.

We thank the reviewer for his/her insightful suggestions. We have now made a substantial effort to investigate mitochondrial function in the frozen tissue samples from the experiments already described in the manuscript. We have modified the recently published Respirometry In Frozen Samples (RIFS) -protocol by Acin-Perez et al. (PMID: 32432379) for Oroboros O2k respirometer to quantify CI, CI&CII and CII mediated mitochondrial respiration. Acin-Perez et al. have demonstrated that the electron transport chain (ETC) remains intact in mitochondrial inner membrane upon freezing and this RIFS-protocol allows the measurement of ETC activity despite the disruption of ATP-coupled respiration. Our modified protocol has now been validated using several sample types with a substantial number of experiments. We have observed similarly than Acin-Perez et al. that changes in mitochondrial respiration, detected in fresh tissue samples, can be also quantified in frozen tissues with the RIFS protocol. We have this method-related manuscript under preparation and it will be submitted to Biorxiv and mitochondria-related journal during the coming spring.

As can be seen below in Fig. 3 and in the manuscript in Fig. 4a,j, we noticed that CI, CI&CII and CII mediated mitochondrial respiration was significantly decreased in skeletal muscle of C26-F and VCM mice in comparison to control mice; niacin treatment did not significantly affect these measured parameters. In the liver (Fig. 3 below and manuscript Fig. 5b, h), CI, CI&CII and CII mediated mitochondrial respiration tended to be enhanced in C26-F mice, again with no niacin effect. In contrast, VCM demonstrated significantly reduced hepatic mitochondrial respiration which was normalized to the level of control mice by niacin.

UNIVERSITA' DEGLI STUDI DI TORINO
DIPARTIMENTO DI SCIENZE CLINICHE E BIOLOGICHE

Unità di Medicina Sperimentale e Patologia Clinica – C.so Raffaello 30, 10125 Torino

Fig. 3. a-d Mitochondrial respiration as O₂ flux normalized by the tissue weight in muscle (a, c) and liver (b,d) used in the assessment (mg) in C26-F, VCM and control mice ($n=6-8$). NA, niacin.

Furthermore, we performed biochemical analysis of mitochondrial respiratory chain complex I-IV and citrate synthase enzymatic activities in the skeletal muscle of C26-F mice, please see below in Fig. 4, manuscript Fig. 4d and Supplementary Fig. 3g-j. No significant changes were detected in comparison to control mice but there was a trend for increased complex II enzymatic activity after niacin treatment.

It should be mentioned that NA-treatment mediated robust increase in ATP content in C26-F skeletal muscles (please see next comment, Fig. 5) suggesting that NA may have an impact on OXPHOS by increasing ATP synthase activity even if only mild increase in mitochondrial function was observed based on complex II enzymatic activity results. Based on these new data and together with previous results on PINK1, we also suggest that niacin may improve muscle mitochondrial membrane potential in C26-F mice, leading to normal activity of ATP synthase and the partial restoration of ATP level (please see manuscript Fig. 4b,i) and the discussion (lines 337-346).

UNIVERSITA' DEGLI STUDI DI TORINO
DIPARTIMENTO DI SCIENZE CLINICHE E BIOLOGICHE

Unità di Medicina Sperimentale e Patologia Clinica – C.so Raffaello 30, 10125 Torino

Fig. 4. a Citrate synthase (CS) activity measured by substrate consumption over time (nmol/min) normalized by the total amount of protein (mg) in the lysate in C26-F and control mice ($n=7$). b-e Enzymatic activity of mitochondrial complexes (CI, II, III and IV) measured by substrate consumption/product generation over time (nmol/min) normalized by the total amount of protein (mg) in the lysate in C26-F and control mice ($n=5$). NA, niacin.

4. *If reduced muscle NAD⁺ in C26-F model is associated with impaired mitochondria as claimed by this manuscript, what is energy production or ATP level in control, C26-F and Niacin treatment groups?*

The reviewer is raising a relevant point. We have now measured ATP content in skeletal muscle of C26-F and VCM mice. As can be seen below in Fig. 5 and in manuscript Fig. 4b, NAD⁺ loss and mitochondrial dysfunction were associated with lower ATP content in C26-F mice in comparison to control mice. We are particularly grateful to the reviewer for asking this additional measurement since we found that niacin partially restored skeletal muscle ATP content in this mouse model, providing further evidence of niacin's effectiveness in rescuing muscle bioenergetics. Consistent with the unaffected muscle NAD⁺, no significant changes in ATP content was detected before or after niacin treatment in the skeletal muscle of VCM mice (Fig. 5 and in manuscript Fig. 4k). It is worth mentioning that in the last months we could slightly increase the VCM animal cohort up to 9 mice per group. Despite no changes in muscle ATP levels occurred in VCM untreated mice, confirming that the VCM represents a mild cancer cachexia model, the increased n in the niacin-treated mice allowed to measure a borderline significant increase, confirming that boosting NAD⁺ metabolism has a positive impact on muscle energy availability. In order to avoid confounding the reader, we decided to keep in the manuscript the original animal cohort that has been used for all the measurements reported, showing the results on the enlarged cohort only here for better detailing the work undergone for the revision and better circumstantiate our conclusions.

Fig. 5. ATP concentration (μM) normalized to protein amount (mg) in C26-F, VCM and control mice ($n=7-9$). NA, niacin.

In summary, the observed intracellular loss of ATP content in C26-F mice suggests that this model exhibits severe energy metabolism failure in skeletal muscle which can be partially ameliorated by niacin. The niacin-promoted increase in ATP content is also a proof of a metabolic condition permissive for improvements in C26-F mice. As expected, VCM mice with a mild chronic cachexia did not present alterations in ATP content, although being responsive to niacin administration.

Reviewer #2 (Remarks to the Author); expert in cancer-associated cachexia:

Beltra, Pöllänen, and colleagues present a manuscript describing a role for niacin (NA) supplementation in reducing experimental cancer cachexia. This work follows their previous work, which demonstrated in reduced NAD⁺ and reduced Nr2 in colon-26 (C-26) tumor-bearing mice. In this manuscript, the authors describe similar results in C-26 mice treated with folfox (C-26F), and also reduced NAD⁺ in KPC pancreatic cancer mice. The authors report that niacin (NA) supplementation reduces muscle wasting in C-26F and VCM mice, despite the fact that NAD⁺ levels are not significantly reduced in VCM mice, and suggest that NA may be sufficient to improve energy metabolism in cancer patients to prevent cachexia. They support this claim with data demonstrating clear build-up of citrate in the TCA cycle and glycolysis products in muscle from cancer patients with low Nr2 compared to healthy individuals or cancer patients with high Nr2, indicating clear changes in metabolism associated with Nr2 gene expression.

We are also grateful to this reviewer for highlighting the most relevant positive findings of our research. It is nice to observe that distinct reviewers focus most of the attention on slightly different aspects of the investigation, allowing us to provide an improved version of the manuscript that hopefully will better suit the journal's readers.

Major concerns

UNIVERSITA' DEGLI STUDI DI TORINO
DIPARTIMENTO DI SCIENZE CLINICHE E BIOLOGICHE

Unità di Medicina Sperimentale e Patologia Clinica – C.so Raffaello 30, 10125 Torino

5a Because the difference in tumor burden is not statistically significant does not mean that it is not physiologically meaningful (supplemental Figure 2h). While the error bars are large, the mean difference is perhaps 30% of the untreated tumor size, which is likely physiologically meaningful.

We thank the reviewer for this valuable comment and we agree this is a critical point we consider in every intervention study in tumor-bearing mice. Even if we already provided an initial in vitro screening of niacin effect on C26 cells (manuscript Supplementary Fig. 2f,g) that is in contrast with the idea of niacin impairing tumor growth, we understand that the reviewer is pointing to the apparently lower tumor mass in NA-treated C26-F mice as a potential tumor intrinsic niacin action that can explain the reduced cachexia resulting from impaired tumor growth. We are here reporting additional data that further rebut this hypothesis and hopefully will convince the reviewer about the mostly tumor extrinsic rather than tumor intrinsic niacin action.

If niacin impairs tumor growth, this would turn into a prolonged animal survival. This is not the case, as shown below in Fig. 6 and in manuscript Supplementary Fig. 2k, suggesting that improved cachexia is not affecting tumor progression leading to animal death at comparable endpoints. For this experiment, young adult (3 months old) animals were used due to the limited availability of adult/middle-aged mice, however we do not foresee relevant changes in 6-month-old animals as used for the main experiment.

Fig. 6. Kaplan-Meier plot showing fraction of surviving mice in each time point after tumor inoculation. Dotted lines indicate Folfox treatment. Statistical analysis was performed with the Mantel-Cox test (C26-F $n=12$, C26-F + NA $n=11$). NA, niacin.

5b The fact that nicotinamide riboside was unable to rescue cancer cachexia increases concerns that the protection in muscle mass is due to tumor-effects, not muscle effects,

Nicotinamide riboside (NR) has actually been very recently published to alleviate cancer cachexia but NAD⁺ metabolism was not properly investigated in this study (PMID: 34262449, ref. 10 in the manuscript). The circulating TNF- α and IL-6 levels were significantly decreased upon NR in this study, but the authors did not demonstrate whether NR affected the production of these cytokines by the host or the tumor. Therefore, it remains unclear whether NR's protective effect was due to tumor-effects, muscle/host systemic effects or both in this study.

UNIVERSITA' DEGLI STUDI DI TORINO
DIPARTIMENTO DI SCIENZE CLINICHE E BIOLOGICHE

Unità di Medicina Sperimentale e Patologia Clinica – C.so Raffaello 30, 10125 Torino

5c and this concern is compounded by the ability of NA to afford significant protections to VCM mice without the expected changes in NAD+-associated markers.

We thank the reviewer for the comment, however, we only partly agree with this statement. Niacin affords protection to VCM mice in the absence of muscle NAD+ loss. This is not conclusive on the lack of NAD+ metabolism impairment in this model and the impact of niacin administration, since:

- muscle *Nrk2* transcript and protein levels are reduced (manuscript Fig. 1);
- muscle mitochondrial respiration and content is impaired (manuscript Fig. 4);
- liver NAD+ levels are reduced, along with mitochondrial respiration and content (manuscript Fig. 5);
- Niacin rescues some of the alterations while does not protect from others; we would expect an overall protective effect in case of reduced tumor growth;
- Niacin was given only for the final 4 experimental weeks in mice that present with the genetic alteration leading to spontaneous tumor development since birth (13 months earlier).

5d Tumor-intrinsic versus tumor-extrinsic effects are important to appreciate because the likely therapeutic potential is higher with tumor-extrinsic effects.

We fully agree with the reviewer and we hope we here provided evidence of tumor-extrinsic niacin effects, even if we do not fully discard the idea that considering cancer as a whole body disease, niacin could potentially provide benefits even in directly fighting the tumor by improving anti-cancer therapies, for example by boosting the effects of tumor immunotherapy in parallel to improving cachexia. Future investigations will clarify this point.

5e To demonstrate true independence from tumor-burden, the relationship between muscle weight and tumor burden should be assessed for linearity, similar to PMID 32407241.

We performed the analysis as suggested by the reviewer (see below Fig. 7). Looking at the numbers, the only significant relationship occurred for the *gastrocnemius* (GSN; P=0.05) in C26-F mice receiving niacin (Fig. 7b), while the other 3 tests resulted in non-significant relationships. This situation leaves room for interpretation and speculation, being neither in favor of the reviewer's nor of the author's position. One neutral comment is that this confusion may be due to the inappropriateness of tumor mass assessment as a proxy of cancer cell burden, being the tumor mass composed by a mixture of cancer cells, stroma and necrotic tissue, all three well represented and impeding to extrapolate a clear indication of cancer cellularity. Beyond the numbers, we dare to present here one consideration looking at the graphs. For both *tibialis anterior* (TA) and GSN most samples are clustered according to the treatment, while for tumor mass the treatment does not allow to discriminate between the two groups, even if 3 mice in the niacin-treated group are on the lower tumor size side, likely being the cause of this discussion. Overall, the relationship between muscle weight and tumor burden is not clear in our study groups.

UNIVERSITA' DEGLI STUDI DI TORINO
DIPARTIMENTO DI SCIENZE CLINICHE E BIOLOGICHE

Unità di Medicina Sperimentale e Patologia Clinica – C.so Raffaello 30, 10125 Torino

Fig. 7. a-b Correlation plot between tumor weight and either *tibialis anterior* (TA) (a) or *gastrocnemius* (GSN) (b) muscle weight of individual C26-F mice ($n=7$). Statistical analysis was performed with Pearson's test. NA, niacin.

5f Demonstrating that circulating myostatin or IL-6 do not change with NA treatment would further support the idea that the effects of NA are tumor-independent.

We thank the reviewer for this very good comment. Both myostatin and IL-6 are also produced by the host, making it hard to dissect in the serum the contribution of the tumor from that of the host. In order to circumvent this issue, we now extracted the mRNA from the tumors (C26-F experiment) and measured IL-6 and activin a transcripts (*il6* and *inhba* genes, manuscript Supplementary Fig. 2j and below Fig. 8), since myostatin transcript levels were very low and in some samples below the detection limit. Further, activin a better associated than myostatin with cachexia and patient survival (PMID: 25751105 and 28712119). The results show no difference or even a tendency to increase in niacin-treated animals, suggesting that the cytokine production from the tumor (either cancer or stromal cells) is likely comparable and that the tendency towards smaller tumors may not reflect a lower cancer cell burden.

Fig. 8. Relative expression of *Il6* and *Inhba* genes in the tumors of C26-F and C26-F + NA groups ($n=7$). Statistical analysis was performed either with Student's t-test or ANOVA + Fisher's LSD. NA, niacin.

UNIVERSITA' DEGLI STUDI DI TORINO
DIPARTIMENTO DI SCIENZE CLINICHE E BIOLOGICHE

Unità di Medicina Sperimentale e Patologia Clinica – C.so Raffaello 30, 10125 Torino

Regarding the systemic host inflammatory response to tumor growth, we here report an extract of the results from a side project running in collaboration with prof. Shinpei Kawaoka (Kyoto University) in which the liver transcriptome has been sequenced ($n=4$ for each experimental group). The liver contributes to systemic inflammation by releasing acute phase reactants (APRs), as we and others previously showed. The two most relevant APRs in tumor-bearing mice are serum amyloid A (*Saa1*) protein and *Serpina3n*. In Table 2 you can find the gene expression (TPM scores) in the three mouse groups of the C26-F and VCM experiments of the two APRs along with *Il6* and *Inhba* as comparison for the previously presented ‘tumor-related’ genes. It is clear again that the acute C26-F model is characterized by a more severe hepatic inflammation as compared to the chronic and milder VCM model. In both cases niacin administration does not interfere with host response to tumor presence, corroborating again the concept that niacin effects are likely to be ascribed to the regulation of host metabolism rather than to a either direct or indirect action on the tumor.

	C	C26-F	C26-F + NA
Saa1	170	35816	45605
Serpina3n	394	5487	5519
il6	<1	<1	<1
inhba	38	23	39

	C	VCM	VCM + NA
Saa1	758	3465	8875
Serpina3n	367	2312	567
il6	<1	<1	<1
inhba	25	35	40

Table 2. Relative abundance of transcripts associated with systemic host inflammatory response in the livers of C26-F (upper) and VCM (lower) experiments ($n=4$). NA, niacin.

5g Further, if possible, differences in tumor burden should be assessed in the VCM animals.

The VCM model is characterized by spontaneous tumors that are heterogeneous in terms of morphology, size, localization along the gut (from duodenum to colon) and number (in rare cases we found 2 tumor masses, please see Fig. 9), making it hard to perform any kind of comparison based on the primary mass. This condition can be compared to the human syndrome, where the occurrence of cachexia is independent from tumor size, and strongly dependent from host systemic inflammation, that has been discussed in the previous points. We are sorry for not having recorded the tumor mass from the experiment here presented, however, our initial characterization of the model suggested that primary tumor mass has not an analytical significance for the study. Again, we would like to remind that the mice were left untreated for 12 months to allow the tumors to spontaneously develop and niacin was administered only for the following 4 weeks, making unlikely that an interference with tumor development occurred. By the way, we clearly stated the lack of information on tumor burden as a limitation of this study (results, lines 176-177).

UNIVERSITA' DEGLI STUDI DI TORINO
DIPARTIMENTO DI SCIENZE CLINICHE E BIOLOGICHE

Unità di Medicina Sperimentale e Patologia Clinica – C.so Raffaello 30, 10125 Torino

Fig. 9. a-c Representative pictures of the intestine of 12-month-old VCM mice ($n=3$). Red circles highlight detected tumor masses.

6 *The increase in serum ATP in low NRK2 patients is perplexing to me, and is somewhat confusing, as I would expect circulating energy levels to be similar to muscle ATP levels. Is the hypothesis that there is more circulating ATP because these patients are required to make ATP outside of their muscles, due to the TCA cycle defect? At a minimum, this would be a valuable discussion point.*

We thank the reviewer for the care in checking the results and for the ability to notice such inconsistency that was due to our mistake. The reviewer is right in reporting that in the previously submitted manuscript version, serum ATP levels were shown to be increased in patients with low *Nrk2* muscle expression. There are two main issues with this result, the first regarding the significance and a second technical one. The first is that extracellular ATP, a very unstable metabolite, does not reflect muscle intracellular ATP, especially in the context of cancer patients. Indeed, circulating ATP is a potential marker of tissue stress or damage (PMID: 32280302 and 33212982). The second, we apologize for this, is that after a recheck of the data provided by the metabolomics facility, we understood that all the circulating nucleotide signals were, unlike in the muscle, below the limit of detection considered acceptable by the facility. In the current revised version of the manuscript that we are submitting, the nucleotides that were under the detection limits have been removed.

7a *The heterogeneity of patient *Nrk2* expression is not particularly surprising (Figure 2a), but is there an opportunity to learn from it, perhaps by associating *Nrk2* expression to metabolite concentrations?*

We thank the reviewer for this suggestion. We have now included two correlation graphs (i.e. those showing statistically significant results) regarding fructose-6-phosphate and citrate concentrations and *NRK2* expression in manuscript Fig. 2f,g and below Fig. 10. The correlation graphs with other metabolites not showing statistically significant results are presented in manuscript Supplementary Fig. 2. This evidence that

UNIVERSITA' DEGLI STUDI DI TORINO
DIPARTIMENTO DI SCIENZE CLINICHE E BIOLOGICHE

Unità di Medicina Sperimentale e Patologia Clinica – C.so Raffaello 30, 10125 Torino

Nrk2 loss correlates with substrate build-up is further supporting the concept of impaired metabolite utilization leading to energy failure (see comment to reviewer 1, point 2).

Fig. 10. a-b Correlation plot between *NRK2* gene expression and fructose-6-phosphate (F6P; a) or citrate (b) abundance in individual cancer patients (further information in the manuscript). Statistical analysis was performed with Spearman's test ($n=20$).

7b Similarly, there is significant heterogeneity of weight loss in the low *NRK2* patients utilized in Figure 2c. Is there an opportunity to leverage this heterogeneity and link high weight loss to NAD^+ levels?

To accommodate this question we tested the linearity between weight loss and muscle NAD^+ levels. As presented here (see below in Fig. 11) and in manuscript Supplementary Fig. 1g, no correlation exists between these two variables. We believe, as discussed in the manuscript (second paragraph of the discussion), that body weight loss is not sufficiently informative about impaired muscle metabolic function.

Fig. 11. Correlation plot between body weight loss (WL) and muscle NAD^+ levels of individual cancer patients. Statistical analysis was performed with Spearman's test ($n=20$).

8 It is intriguing that both *NRK2* high and *NRK2* low patient muscles demonstrated lower NAD^+ content than muscle from healthy individuals (Figure 2f). This would make for an interesting discussion point.

UNIVERSITA' DEGLI STUDI DI TORINO
DIPARTIMENTO DI SCIENZE CLINICHE E BIOLOGICHE

Unità di Medicina Sperimentale e Patologia Clinica – C.so Raffaello 30, 10125 Torino

Changes in intracellular NAD⁺ levels are typically caused by alterations in NAD⁺ biosynthesis, activities of NAD⁺ degrading enzymes, metabolic reactions relying on NAD⁺/NADH redox couple or a combination of all these processes. Thus, the reason for similar NAD⁺ content in NRK2 high and NRK2 low patients could be complex. It is possible that impaired glycolysis (as suggested by the accumulation of glycolysis intermediates) elevates NAD⁺ levels (consumed in the glycolysis) and in combination with decreased NRK2-mediated NAD⁺ biosynthesis, this may result in similar NAD⁺ levels in NRK2 low patients in comparison to NRK2 high patients. However, as we do not have detailed data about NAD⁺ metabolism in these patients, we prefer to refrain from the speculation of the mechanism. We agree that it would be highly interesting to investigate NAD⁺ metabolism more in detail in these patients in future studies.

9a Niacin seems to be somewhat less effective in the VCM mice, particularly in returning mitochondrial measures to control levels.

We understand the reviewer refers mainly to the results presented in manuscript Fig. 4. In the revised manuscript, Fig. 4 includes additional data (respirometry, intracellular ATP, citrate synthase activity). Looking at the picture as a whole, it now appears that there is not a relevant difference in niacin effect between C26-F and VCM mice. An alternative interpretation is based on the distinct severity of these two models, however we are not willing to speculate on each single result based on muscle atrophy or NAD⁺ depletion, thinking that is the overall signature that should guide the interpretation of the results.

9b Is the hypothesis that this is because significant weight loss has already occurred prior to 12 months of age in these mice? Only part of the cachexia is occurring during the treatment window?

We thank the reviewer for the comment that helps us better understand the significance of our dataset. We find the first hypothesis of the reviewer appropriate and reasonable, however it requires a clarification. In this model body weight loss occurs very slowly and is borderline significant at the end of the experiment (manuscript Supplementary Fig. 1a), while muscle atrophy is overt (manuscript Supplementary Fig. 1d). Monitoring the body weight before starting niacin administration shows a significant body weight loss at 11 months of age (see below in Fig. 12), that is compatible with the presence of muscle atrophy when the treatment started. It is worth mentioning that even the control mice show a progressive decline of body weight associated with the progression of aging, making it hard to distinguish between age-related and cancer-related sarcopenia. In any case, the aim of this research is to provide evidence of therapeutic action, i.e. the ability to treat, not to prevent, and niacin works well in rescuing sarcopenia in this experimental setting. We skip here the speculations linking the molecular profile to the macroscopic cachexia features that have been extensively discussed in the above points.

UNIVERSITA' DEGLI STUDI DI TORINO
DIPARTIMENTO DI SCIENZE CLINICHE E BIOLOGICHE

Unità di Medicina Sperimentale e Patologia Clinica – C.so Raffaello 30, 10125 Torino

Fig 12. Body weight changes (%) in VCM and control mice according to age prior to niacin treatment (months 7-11: control and VCM $n=8$; months 11.5-12: control $n=7$, VCM $n=5$). Statistical analysis was performed using Student's t-test.

9c Because NAD⁺ actually needs to be decreased for niacin supplementation to be effective (Figure 1a)?

In our opinion, the complexity of NAD⁺ metabolism does not allow to formulate such a conclusive statement (see also point 8). However, the most clinically relevant cachexia parameter, sarcopenia/muscle atrophy, was counteracted by niacin in the VCM model, highlighting the relevance of systemic NAD⁺ metabolism dysregulation due to both muscle *Nrk2* loss and liver NAD⁺ depletion. In humans, niacin has been shown to be effective and elevate systemic NAD⁺ levels and improve muscle strength even in healthy controls without blood or muscle NAD⁺ depletion (PMID: 32386566).

Overall, in our opinion, the differences reported should be ascribed to the peculiarities of the models, C26-F being acute and severe while VCM mild and chronic, the latter closer to the human scenario.

Minor concerns

10 The language surrounding the depletion of NAD⁺ in "severe" cachexia (line 92) is likely overly strong, as the VCM mice showed much lower muscle masses than their controls, but did not display reduced NAD⁺.

We kindly point out that the reviewer has probably misunderstood this sentence. What we meant by this sentence was that NAD⁺ depletion is observed in the mouse models with severe cachexia, i.e. C26-F and KPC mice, not in VCM mice with mild cachexia (compared to the C26-F model, see below in Fig.13 for direct comparison).

Fig. 13. Loss of *tibialis anterior* (TA) and *gastrocnemius* (GSN) muscles mass in C26-F (n=7) and VCM (n=8) mice.

11 Additional clarity would be appreciated on the method of normalization for *Nrk2* gene expression from patient samples and the conversion to AU.

NRK2 gene expression results are normalized to housekeeping gene expression and displayed as fold change over control group (healthy patients) as better specified in revised manuscript Fig. 2a and 2b and in materials and methods section (line 131 and 482, respectively).

12 Further, while the methods are clear that *Nrk2* gene expression, not protein expression, was measured in muscle from cancer patients, the results could be somewhat more clear.

We have better specified in the results section that *NRK2* gene expression was evaluated in the patient's muscles (lines 109, 111, 115).

13 Technically it is a trend towards further downregulation of *Nrk2* in cachectic patients compared to pre-cachectic patients (line 109, Figure 2a).

We have corrected the text accordingly (actual line 112).

14 While I agree that Fig. 2e-h are consistent with altered energy metabolism (line 118), calling these data evidence of a hypercatabolic state, which technically indicates protein breakdown, is not supported. Similarly, hypercatabolism is not supported in line 120.

We rephrased the first sentence, suggesting only a potential link to hypercatabolism, since no direct evidence is reported in this manuscript. The other way around muscle protein breakdown leads to increased muscle amino acid accumulation as already demonstrated in animal models where protein hypercatabolism was extensively characterized (see point 2). We agree with the second point, given that the serum amino acid profile is not clearly allowing to trace muscle proteolysis; indeed the comment on hypercatabolism has been removed (actual lines 122-125).

15 The chemotherapy status (treated or not) of patients should be added to supplementary table 1, as should the breakdown of pancreatic and colon cancer patients. Similar information should be added to supplementary table 2.

UNIVERSITA' DEGLI STUDI DI TORINO
DIPARTIMENTO DI SCIENZE CLINICHE E BIOLOGICHE

Unità di Medicina Sperimentale e Patologia Clinica – C.so Raffaello 30, 10125 Torino

The chemotherapy exposure and the tumor type information have been added in the revised manuscript Table S1 (Patient population's characteristics). The revised manuscript Table S2 (Metabolomic patient subset's characteristics) contains the requested information that in the previous version was split.

16 *That C-26F experiments were conducted only in female mice and VCM mice were only in males should be more prominently discussed in the results section, as a sexual dimorphism cannot be excluded.*

We now mentioned this point (the information was already present in the methods) in the results section when introducing the animal models and stressed on the limitation of not being able to exclude sexual dimorphism (lines 79-80).

17 *In the discussion line 320, I am not sure that the lack of data on liver contributing to muscle wasting in the cancer cachexia models used can be used to exclude that NA supplementation improves muscle wasting at least in part through a liver-dependent mechanism, particularly given the results from Figure 5. This sentence could use additional support or revision.*

We agree that the conclusion provided in the sentence was too subjective. We have modified it, being more open to future new investigations on liver alterations in cancer cachexia we are recently starting and that, however, are out of scope in this work (lines 359-363).

On behalf of all the authors,

Fabio Penna
Department of Clinical and Biological Sciences
University of Torino (Italy)

REVIEWERS' COMMENTS

Reviewer #1 (Remarks to the Author):

Most of my previous concerns were addressed or explained in revised manuscript or rebuttal letter.

The revised manuscript added new data to show that skeletal muscle NAD⁺ loss and mitochondrial dysfunction were associated with lower ATP content in C26-F mice in comparison to control, but not in VCM mice.

Reviewer #2 (Remarks to the Author):

Beltra, Pollanen, and colleagues present a revised version of their manuscript describing the ability of niacin supplementation (NA) to maintain muscle mass in multiple models of cancer cachexia. In my opinion, the manuscript is much improved, with significant amounts of new data that clarify and further support the authors' main points. My remaining concerns are generally minor in nature.

Line 82 – I believe that “reduced muscle mass” more accurately describes the lower muscle masses found in cachectic mice compared to controls instead of “muscle wasting” – with the data presented, the authors cannot exclude of mice to add failure to add muscle mass as a cause of these differences.

Figure 2 - Pre-cachectic is not the best term for weight-stable cancer patients, as it is not known that they will develop cachexia.

Line 228 – “modestly improves” would be a more accurate term than “ameliorates skeletal muscle mitochondrial status” – oxygen consumption really does not improve in skeletal muscle.

While it is appreciated that there may not be blood available for analysis and this reviewer acknowledges Table 2 provided in the response to reviewers showing no changes, gene expression of Il6 and inhba should be determined for cachectic livers and added to the manuscript to exclude suppression of IL-6 or activin A synthesis as a mechanism by which NA is working to improve whole body metabolism.

In line 338, I believe that the authors are speaking in hypothetical terms about how NA may maintain skeletal muscle mass and not claiming that this is a mechanism that their data support, but it is somewhat unclear and the text should be edited. Line 343, which refers to slightly improved mitochondrial function, is a more fair characterization of the data presented.

RESPONSE TO REVIEWERS' COMMENTS

Reviewer #1 (Remarks to the Author):

Most of my previous concerns were addressed or explained in revised manuscript or rebuttal letter. The revised manuscript added new data to show that skeletal muscle NAD⁺ loss and mitochondrial dysfunction were associated with lower ATP content in C26-F mice in comparison to control, but not in VCM mice.

We are grateful to the reviewer. His valuable feedback and constructive suggestions have significantly improved the quality of the manuscript and allowed us to produce data to better support our conclusions.

Reviewer #2 (Remarks to the Author):

Beltra, Pollanen, and colleagues present a revised version of their manuscript describing the ability of niacin supplementation (NA) to maintain muscle mass in multiple models of cancer cachexia. In my opinion, the manuscript is much improved, with significant amounts of new data that clarify and further support the authors' main points. My remaining concerns are generally minor in nature.

We thank the reviewer for the appreciation of our efforts and for the new deep analysis of our work in order to further improve it suggesting minor changes that have been well taken and can now be found in the newly revised manuscript. Below, a point-by-point response detailing the last changes is provided.

1 Line 82 – *I believe that “reduced muscle mass” more accurately describes the lower muscle masses found in cachectic mice compared to controls instead of “muscle wasting” – with the data presented, the authors cannot exclude of mice to add failure to add muscle mass as a cause of these differences.*

Following the reviewer's suggestion, we have substituted the term “muscle wasting” for “decreased muscle mass” (line 81).

2 Figure 2 - Pre-cachectic *is not the best term for weight-stable cancer patients, as it is not known that they will develop cachexia.*

We agree with the reviewer, even if weight-stable colorectal and pancreatic cancer patients may be considered pre-cachectic, given the high probability to develop cachexia. Following the suggestion, the term “pre-cachectic” has been replaced by “weight-stable” in the text (lines 104-105), in Fig. 2a and in Supplementary Table 1 (abbreviated as WS).

3 Line 228 – *“modestly improves” would be a more accurate term than “ameliorates skeletal muscle mitochondrial status” – oxygen consumption really does not improve in skeletal muscle.*

We agree with the suggestion and the term “ameliorates” has been replaced by “modestly improves” (line 189).

4 *While it is appreciated that there may not be blood available for analysis and this reviewer acknowledges Table 2 provided in the response to reviewers showing no changes, gene expression of Il6 and inhba should be determined for cachectic livers and added to the manuscript to exclude suppression of IL-6 or activin A synthesis as a mechanism by which NA is working to improve whole body metabolism.*

We completely agree with the reviewer's comment and in order to include the new data suggested within the manuscript, we analyzed the hepatic gene expression of systemic inflammatory markers *Il6*, *Inhba* and *Saa1-2* by qPCR (see below in Fig. 1). Overall, these results confirm the sequencing data presented in **point 5f** of the previous round of review: C26-F mice present a more severe hepatic inflammatory phenotype as compared to the chronic and milder VCM model, with niacin not interfering with these responses. As in the sequencing data, liver samples presented very low expression of *Il6* in all three groups of both models (mean Ct=36 for C26; mean Ct=36 for VCM). These panels are now included in the manuscript as **Supplementary Fig. 3j** (manuscript lines 144-145) and **Supplementary Fig. 4g** (manuscript lines 157-158).

Fig. 1. a-b Relative gene expression of *Il6*, *Inhba* and *Saa1-2* genes in the liver of **(a)** C26-F ($n=7$ per group) and **(b)** VCM ($n=8$ per group, except for VCM *Saa1-2* $n=7$) models. Statistical analysis was performed with a two-sided ANOVA + Fisher's LSD for normally distributed data.

5 In line 338, I believe that the authors are speaking in hypothetical terms about how NA may maintain skeletal muscle mass and not claiming that this is a mechanism that their data support, but it is somewhat unclear and the text should be edited. Line 343, which refers to slightly improved mitochondrial function, is a more fair characterization of the data presented.

We agree with the reviewer that more clarity was needed, indeed now we stated that the concept is based on a speculation and requires further experimental demonstrations. Also the verbs used now point to a more hypothetical interpretation of the results (lines 260-263).